# ROCK regulates the intermittent mode of interstitial T cell migration in inflamed lungs

Paulus Mrass [1], Sreenivasa Rao Oruganti[1], G. Matthew Fricke[2], Justyna Tafoya[2,3], Janie R. Byrum [1],
Lihua Yang[4], Samantha L. Hamilton[4], Mark J. Miller[4], Melanie E. Moses[2,5,6] & Judy L. Cannon [1]

Effector T cell migration through tissues can enable control of infection or mediate inflammatory damage. Nevertheless, the molecular mechanisms that regulate migration of effector T cells within the interstitial space of inflamed lungs are incompletely understood. Here, we show T cell migration in a mouse model of acute lung injury with two-photon imaging of intact lung tissue. Computational analysis indicates that T cells migrate with an intermittent mode, switching between confined and almost straight migration, guided by lung-associated vasculature. Rho-associated protein kinase (ROCK) is required for both high-speed migration and straight motion. By contrast, inhibition of $G\alpha_i$ signaling with pertussis toxin affects speed but not the intermittent migration of lung-infiltrating T cells. Computational modeling shows that an intermittent migration pattern balances both search area and the duration of contacts between T cells and target cells. These data identify that ROCK-dependent intermittent T cell migration regulates tissue-sampling during acute lung injury.

[1] Department of Molecular Genetics and Microbiology, University of New Mexico School of Medicine, MSC 08 4660, 1 University of New Mexico, Albuquerque, NM 87131, USA. [2] Department of Computer Science, University of New Mexico, 1 University of New Mexico, Albuquerque, NM 87131, USA. [3] Department of Mathematics, University of New Mexico, 1 University of New Mexico, Albuquerque, NM 87131, USA. [4] Department of Medicine, Division of Infectious Diseases, Washington University School of Medicine, St Louis, MO 63110, USA. [5] Department of Biology, University of New Mexico, 1 University of New Mexico, Albuquerque, NM 87131, USA. [6] External Faculty, Santa Fe Institute, 1399 Hyde Park Road, Santa Fe, NM 87501, USA. Correspondence and requests for materials should be addressed to J.L.C. (email: JuCannon@salud.unm.edu)

T cells contribute to immune protection against infectious agents and cancer or alternatively can mediate tissue damage in inflammatory settings. Imaging studies have revealed that motility of effector T cells within lymph nodes and sites of inflammation is an important component of an effective immune response[1, 2]. T cells in peripheral tissues are thought to perform mostly "informed" motion guided by environmental cues towards target cells[3]. Nevertheless, the precise molecular mechanisms that regulate migration of effector T cells vary in different tissue contexts.

One important mechanism of effector T cell movement in tissues is chemotactic guidance, which facilitates T cell migration toward infectious foci in the skin and liver[4–6]. For example, expression of CXCR3, the receptor for the chemokines CXCL9 and CXCL10, enables movement of CD8+ effector T cells toward infectious foci[4, 5]. In other contexts, for example in inflamed brain, skin and in tumors, T cells follow structural guidance cues, such as extracellular matrix fibres and the vasculature[7–12]. In the skin, such "contact guidance" of T cells is mediated by integrins[10], whereas in tumors integrin-independent contact guidance has been identified[11]. In the absence of integrin-mediated adhesion, T cells may utilize three-dimensional migration strategies and squeeze through pre-formed channels using "amoeboid" motion along a path of least resistance[13]. In vitro experiments of various cell-types, including T cells, indicate that the cell-intrinsic RhoA-ROCK-myosin II pathway, a regulator of the actomyosin cytoskeleton, enables amoeboid squeezing[14–16]. Two-photon studies have confirmed that inhibition of ROCK or myosin II leads to a moderate reduction of the speed of naive T cells in the lymph node[14, 17]. Nevertheless, the relevance of ROCK during effector T cell migration in inflammatory tissues has not been addressed formally. It also needs to be considered that in some cases, such as in the pancreas of diabetic mice, cytotoxic T lymphocytes (CTL) migrate with apparent randomness, independent of environmental guidance cues[18, 19].

Acute lung injury, in particular its severe form acute respiratory distress syndrome, is a clinical syndrome with high mortality. Currently, treatments are limited to supportive management[20]. The syndrome is initiated by an "exudative phase", which is characterized by a massive influx of immune cells, including T cells[20–22]. Data also indicate that effector T cells contribute actively to the progression and resolution of acute lung injury[22, 23]. In particular, experimental and clinical studies have established a link between lung injury and the accumulation of resident CD8+ T cells[24–27]. Even though it is likely that efficient lung tissue-infiltration by CD8+ T cells is important during pathogenesis, interstitial T cell migration during acute lung injury is barely investigated. Although two-photon studies have shown that lung-infiltrating T cells perform active interstitial migration during infection and asthma, we know very little about the molecular mechanisms that enable tissue-navigation of lung-infiltrating T cells[28–31]. A better understanding of the mechanisms that enable efficient lung-infiltration by T cells could be crucial for the development of improved therapies for acute lung injury and other lung diseases.

In the present study, we perform two-photon imaging on mouse lungs during acute lung injury to observe CD8+ T cells during the effector phase of an immune response. We find that CD8+ effector T cells extravasate effectively into the interstitial lung space and then show intermittent motion, switching between confinement and straight migration. Movement along lung-associated vasculature facilitates straight motion. Chemokines fine-tune the speed of lung-infiltrating T cells, but have a marginal effect on intermittent migration. ROCK on the other hand is crucial for T cells to achieve high speeds and straight migration. These data suggest that environmental and cell-intrinsic signals cooperate to enable effective contact-guided tissue-navigation of T cells during acute lung injury.

## Results

**Interstitial T cell migration during acute lung injury.** The principal goal of this study was to analyze the in situ behavior of effector T cells in intact lung tissue. To this end, we used a well-established murine model of acute lung injury and injected the TLR4 ligand lipopolysaccharide (LPS) intranasally into C57BL/6 mice[21, 32]. In vitro activated polyclonal effector CD8+ T cells from Ubiquitin-GFP animals were introduced one day later via intravenous injection. Lungs from recipient mice were analyzed two to five days after adoptive transfer.

We then assessed into which lung compartments the adoptively transferred T cells migrated, and first focused on whether effector CD8+ T cells extravasated into the lung interstitium, a major limiting step of the effector phase of an adaptive immune response. To quantify the percentage of T cells that have reached the interstitial space of the lung, we used a well-established method where intravascular T cells are labeled via intravenous injection of an anti-CD3 antibody shortly before organ harvest, followed by flow cytometry analysis of digested lungs[33]. Analysis of peripheral lymph nodes, where T cells are segregated from the vasculature, revealed that most adoptively transferred GFP+ T cells were negative for the intravascular CD3 stain (CD3−), which indicated that they were extravascular (Fig. 1a, *left panel*). In lungs from healthy control mice, adoptively transferred effector T cells were mostly CD3+, i.e., intravascular, indicating that these cells reach the lung vasculature, yet fail to enter the interstitial space (Fig. 1a, *middle panel*). In contrast, analysis of lungs from mice that received intranasal LPS revealed that the majority of transferred effector CD8+ T cells were CD3−, i.e., extravascular (Fig. 1a, *right panel*). Staining with an anti-CD8 antibody confirmed that typically >90% of adoptively transferred cells were CD8+ (Fig. 1a). Using immunofluorescence, we found that most adoptively transferred GFP+ T cells were solitary in healthy control lungs, but after treatment with LPS, regions with high T cell density emerged (Fig. 1b, *middle* and *right panels*). Consistent with flow cytometry, most T cells in inflamed lungs were negative for intravascular CD3 stain (Fig. 1b, *left* and *right* panels). We then combined intravascular antibody staining with intranasal injection of an anti-CD45 antibody, as reported previously[34], which revealed that ~80% of the adoptively transferred T cells were protected from staining with either antibody (Fig. 1c). The fact that most adoptively transferred T cells were spatially segregated from the vasculature and airways confirmed that during acute lung injury, the majority of effector CD8+ T cells reside in the interstitial lung space.

To experimentally characterize T cell migration within the interstitial space of inflamed lung tissue, we used two-photon imaging of intact explanted inflamed lung tissue (Fig. 1d; Supplementary Movie 1). Analysis of three-dimensional time-lapse sequences revealed specific regions where T cells displayed high migratory activity and other regions with less migration, similar to previous observations in lung tumors[11]. We focused on regions of high migration for the remainder of the study. We were able to image and track effector T cells for up to 3 h and found that T cells performed displacements of up to several hundred micrometers over the observation period (Fig. 1d). Altogether, two-photon imaging showed that during acute lung injury CD8+ effector T cells move actively within the interstitial space of the lung.

**Lung infiltrating T cells move over a wide range of speeds.** In an initial analysis of T cell motion, we found that during the 15 min time frame typically used to analyze T cell movement in live

tissues, T cells move at a median speed of 2.3 μm min$^{-1}$ (Fig. 2a, *green solid line*). This is similar to speeds reported for T cell motility in peripheral sites, including skin, brain and lungs, but slower than the speed seen in lymph nodes[4, 12, 31, 35–37]. In the 15 min time window, we saw that a significant proportion of T cells showed no movement (0–1 μm min$^{-1}$) while others moved at speeds of >7 μm min$^{-1}$. T cells moved with an even greater range of speeds when we determined the "instant" frame-to-frame

speeds (90 s). A higher fraction of T cells showed no movement and some cells displayed speeds close to 10 μm min$^{-1}$ (Fig. 2a, *pink bars*). Conversely, at much longer time periods of up to 2 h, we found that T cell speeds were more homogenous, with most T cells moving between 2 and 4 μm min$^{-1}$ (Fig. 2a, *blue dotted line*). This suggested that lung-infiltrating T cells switch repetitively between fast and slow migration, reminiscent of previous reports of T cells in lymph nodes and infected brain[37, 38].

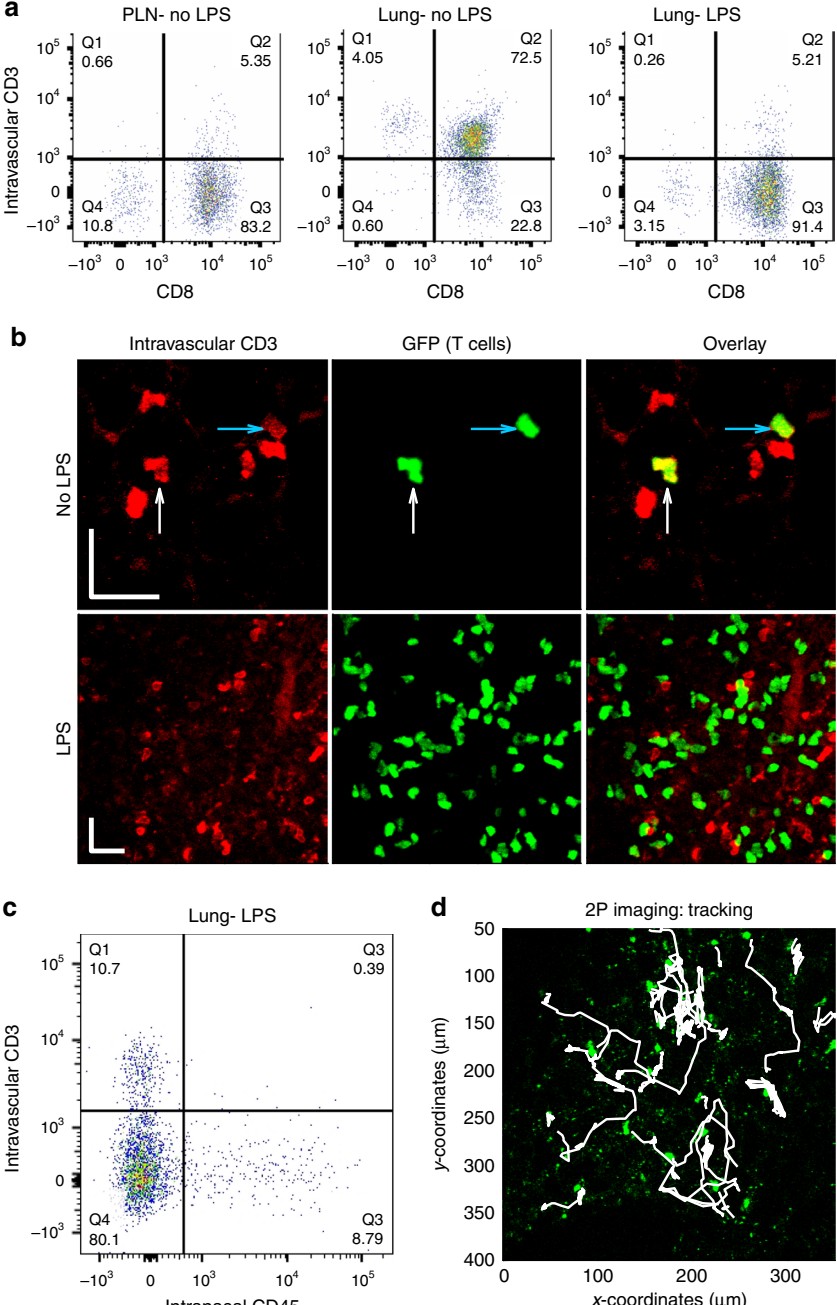

**Fig. 1** Effector T cells localize to the interstitium of the lung upon inflammation. GFP$^+$ effector T cells were adoptively transferred into control and LPS-treated mice and analyzed two to five days later. **a**, **b** Anti-CD3 antibody was injected i.v., followed 5 min later by harvest of peripheral lymph nodes (PLN) and lungs. **a** Single-cell suspensions from the harvested organs were stained with an anti-CD8 antibody and analyzed with flow cytometry. We gated for GFP$^+$ (adoptively transferred) T cells and show the percentage of extravascular (CD3$^-$) and CD8$^+$ T cells. **b** Lungs were fixed with 4% paraformaldehyde and analyzed with confocal microscopy, enabling co-visualization of GFP$^+$ T cells (*green*) and the CD3 surface stain (*red*). *Arrows* highlight double-positive cells. *Scale bars*: 20 μm. **c** Anesthetized mice were injected i.n. with anti-CD45 antibody. After a few minutes, anti-CD3 antibody was injected i.v., followed 5 min later by harvest of lungs. **d** We performed two-photon (2P) imaging of inflammatory regions of explanted lungs. T cells (*green*) and T cell trajectories (*white*) were superimposed on the image. Experiments were repeated three times

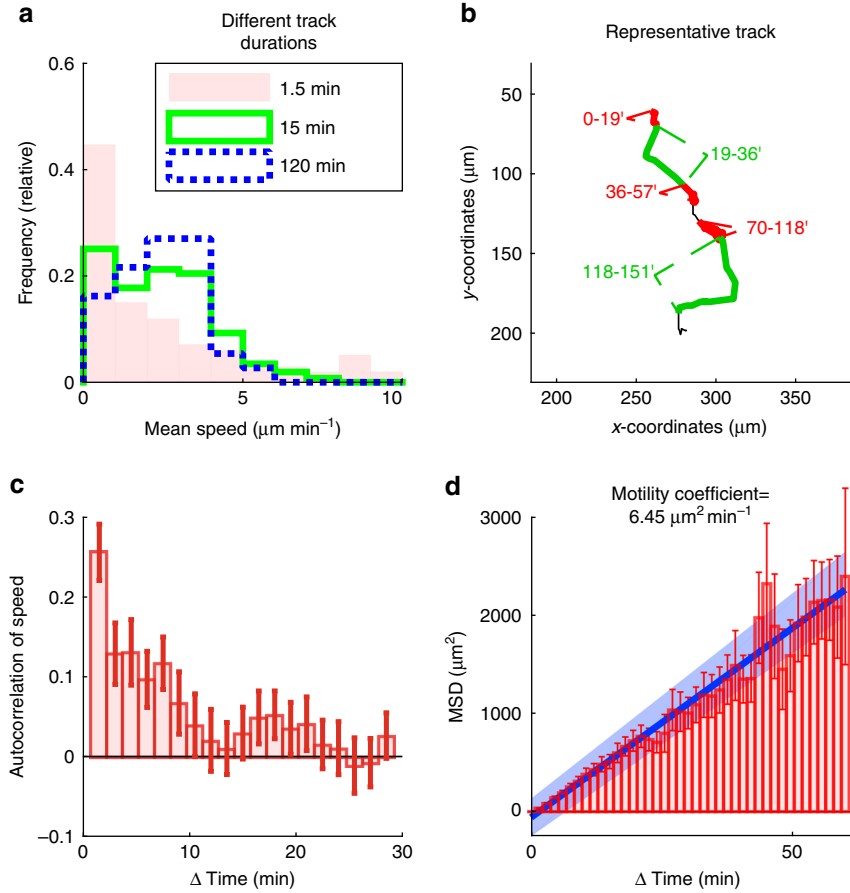

**Fig. 2** Lung infiltrating T cells move over a wide range of speeds. GFP[+] effector T cells were injected i.v. into LPS-treated mice. After two to five days, lungs were explanted and imaged with two-photon microscopy. Tracks with a duration of 120 min were analyzed. **a** Shown are the average speeds of complete 120 min "parent" tracks ($n = 37$ tracks), and of 15 min ($n = 259$ tracks) and 1.5 min ($n = 2923$ tracks) track "children". **b** Representative T cell track with multiple transitions between "fast" (*green segments*) and "slow" migration (*red segments*). Numbers indicate time in minutes. **c** Autocorrelation of speeds. Means of 37 tracks (+/− standard error of the mean). **d** The mean squared displacements (MSD) of 37 tracks (+/− standard error of the mean). The *blue line* and *shaded area* show the linear regression with confidence interval. All quantitative analyses were performed on pooled data obtained from three independent experiments

Indeed, visual inspection of individual tracks revealed that T cells sometimes did not move at all for tens of minutes after which they switched to periods where they moved persistently with high speed (Fig. 2b, compare *green* and *red* segments; Supplementary Movie 2). This was further supported by speed autocorrelation analysis, which revealed a positive correlation for up to several minutes (Fig. 2c). We then plotted the relationship between time and mean squared displacement (MSD). The relationship was roughly linear, and the slope of the curve, estimated by linear regression, revealed a motility coefficient of 6.45 $\mu m^2 min^{-1}$ (Fig. 2d). This motility coefficient is similar to that of memory T cells in the skin but significantly below observations of naive T cells in lymph nodes[15, 39, 40], suggesting that the lung-environment is not particularly conducive to effective tissue-navigation. Taken together, we concluded that lung-infiltrating T cells display an intermittent migration pattern, characterized by repetitive inter-conversion between fast and slow migration.

**Lung T cells show a lognormal correlated random walk.** To explore the migration patterns of lung-infiltrating T cells in more depth, we tested whether their migration behavior could be accurately described with standard migration models. Indeed, the identification of suitable mathematical models, such as simple random walks and Levy walks[2, 3, 36, 38, 41], has enabled the quantification of tissue-infiltration and search behavior of T cells within various tissues. Conversely, significant discrepancies between models and empirical data can help identify characteristic or unusual migration patterns.

First, we plotted the cumulative distribution function (CDF) of experimentally observed two-dimensional speeds of frame-to-frame migration steps (Fig. 3a, *solid black line*), and tested whether it could be accurately fit by relevant theoretical distributions. We found that the Rayleigh distribution, which describes two-dimensional displacements of Brownian motion, was a poor fit for the experimental T cell speeds due to a major underrepresentation of low and high speeds (Fig. 3a, *dashed black line*). Conversely the best fit for power-law distributions, characteristic of superdiffusive Levy walks, showed a severe overrepresentation of low and high speeds (Fig. 3a, *dash-pointed blue line*). We then focused on the log-normal distribution, which we previously demonstrated as an appropriate model for the T cell speeds in lymph nodes[36]. We found that the best fit for a log-normal distribution led to a better recapitulation of experimental speeds (Fig. 3a, *dotted green line*; Supplementary Table 1). However, the log-normal distribution failed to represent a small

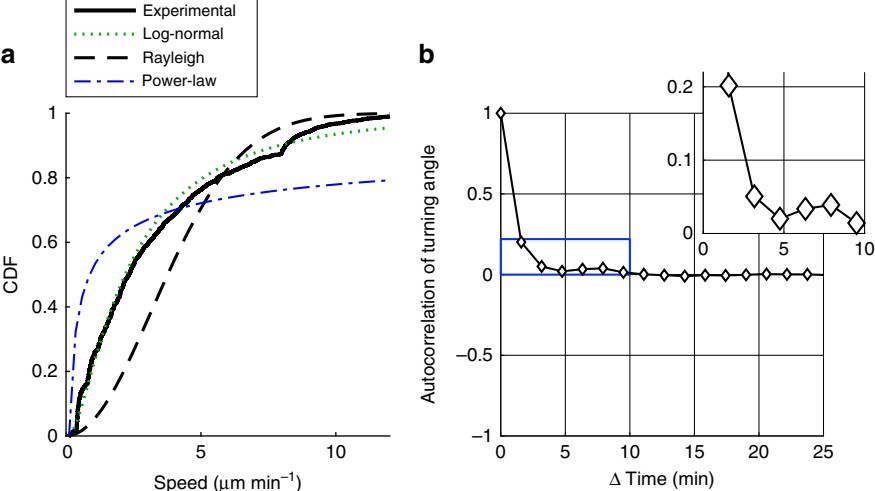

**Fig. 3** Migration of lung-infiltrating T cells is correlated and fits a lognormal distribution. GFP$^+$ effector T cells were injected i.v. into LPS-treated mice. After two to five days, lungs were explanted and imaged with two-photon microscopy. Tracks with a duration of 120 min were analyzed. **a** Shown are the cumulative distribution functions (CDF) of experimentally measured speeds from 2960 frame-to-frame migration steps and of the fits of various mathematical models. **b** Autocorrelation of turning angle. Analyses were performed on 37 tracks pooled from three independent experiments

population of T cells with very high speeds (7.5 to 10 μm min$^{-1}$), reflected by the region to the right of the intersection of the log-normal distribution and the experimental CDF (Fig. 3a). Thus, curve fitting of speeds led to the identification of a small but significant subpopulation of T cells moving at high speeds. Altogether, these data showed that effector T cells in the lung do not move with Brownian or super-diffusive Levy walk strategies.

We also quantified the autocorrelation of turning angles over time. The autocorrelation dropped strongly from 0.2 to 0.05 at time intervals between 1.5 and 3 min, but stayed positive for up to 10 min (Fig. 3b). Thus, T cells maintain some level of directionally persistent migration for several minutes. This also argued against pure Brownian motion, which is characterized by uncorrelated turning angles.

Altogether, the facts that lung-infiltrating T cells have roughly log-normally distributed speeds and show directional persistence indicate that a log-normal correlated random walk is a suitable model of the migration of lung-infiltrating T cells, similar to naive T cells in lymph nodes[36]. Moreover, curve-fitting confirmed the presence of a small subpopulation of high instantaneous speeds, which corroborated that lung-infiltrating T cells switch between different speeds.

**Lung T cells switch between confined and straight migration.** The presence of intermittent high-speed migration in effector T cells prompted us to more carefully characterize the directional behavior of lung-infiltrating T cells. First, we visually inspected motile lung-infiltrating T cells, which revealed that some T cells moved back-and-forth for tens of minutes (Fig. 4a, *magenta segment*; Supplementary Movie 3), and then switched to relatively straight "ballistic" migration (Fig. 4a, *green segment*). To quantify these motility phenotypes, we compared the displacement of original "experimental" T cell tracks with randomized track derivatives. Each randomized track derivative has an identical total length as its experimental source track, but the orientation and order of each directional vector between consecutive time steps was randomized. This allowed a specific comparison of directional persistence while keeping the speed constant (Supplementary Fig. 1). If experimental T cell tracks showed less displacement than randomized track derivatives, the analyzed T cells were considered "confined". Conversely, if the

displacements exceeded that of randomized tracks, T cell migration was considered "straight".

We first analyzed tracks of 15 min duration and generated a single randomized track for each experimental track. Analysis of the displacements revealed that in comparison to the randomized tracks, the experimental T cell tracks were enriched for high and low displacements, while intermediate displacements were reduced (Fig. 4b). This confirmed that lung-infiltrating T cells had a propensity to move either with high directional persistence or to remain confined. To corroborate this finding, we calculated for each experimental and randomized track a straightness Z-score. This score measured how many standard deviations the given displacement deviated from the expected displacement of a randomized track ensemble (Supplementary Fig. 1). Comparison of the straightness Z-scores from experimental and randomized tracks revealed that experimental tracks were enriched for low (confined) and high (straight) straightness Z-scores (Fig. 4c). We confirmed that tracks with high straightness Z-scores at 1 h achieved total displacements of up to several hundred micrometers (Fig. 4d), whereas tracks with low Z-scores showed back-and-forth movement and lower displacements (Fig. 4e).

Given that individual T cells appeared to switch between multiple motility states, we quantified to what degree individual T cells switch between random, straight and confined migration over time. We analyzed 120 min source tracks (80 timesteps) and computed the straightness Z-score of all contained "subtrack children" of 15 min (10 timesteps). Subtracks with Z-scores below −1.7 were defined as confined and subtracks with Z-scores above 1.7 as straight, because only very few randomized tracks had such extreme values (Fig. 4c). Virtually all tracks contained some random subtracks. Strikingly, 16% of the tracks contained a significant number of confined subtracks and 39% of the tracks contained a significant number of straight subtracks. Moreover, 26% of tracks contained both straight and confined subtracks (Fig. 4f). In control tracks, where all subtracks were randomized, fewer than 10% of source tracks contained a significant number of straight or confined subtracks (Supplementary Fig. 2). Altogether, these findings indicated that lung-infiltrating T cells use an intermittent migration strategy and switch between periods of confinement and straight ballistic relocation.

Next, we sought to verify that the observed intermittent migration pattern of lung-infiltrating T cells is not affected by

the lack of blood flow in explanted lung tissue. To this end, we performed intravital two-photon imaging of T cell motion in inflamed lung exposed by surgical preparation, which maintains intact perfusion and ventilation. Visual inspection of lung-infiltrating T cells in live mice revealed transitions between slow, confined and straight migration similar to that seen in explanted lungs (Fig. 4g; Supplementary Movie 4). Moreover, quantitative analysis of motile T cells confirmed fluctuations between low and high speeds (Fig. 4h), as well as between straight and confined migration (Fig. 4i). We conclude that the intermittent migration pattern is a general motility pattern of lung-infiltrating T cells present in explanted tissues and in live mice.

**Contact guidance of lung T cells along the vasculature**. Recent two-photon studies have shown that effector T cells in various inflamed tissues, including skin, neural tissue and tumors, display contact guidance, i.e., move along structural guidance cues such as the vasculature and extracellular matrix fibers[7–12]. We hypothesized that the capacity of T cells to perform directional relocation in lung tissue is dependent on environmental guidance cues within the inflamed lung. We tested whether lung-infiltrating T cells move along the vasculature as observed for T cells in tumors and autoimmune brain lesions[7, 9, 11]. To experimentally address this, we co-visualized GFP+ T cells and vasculature labeled by injection of fluorescently labeled lectin with two-photon microscopy.

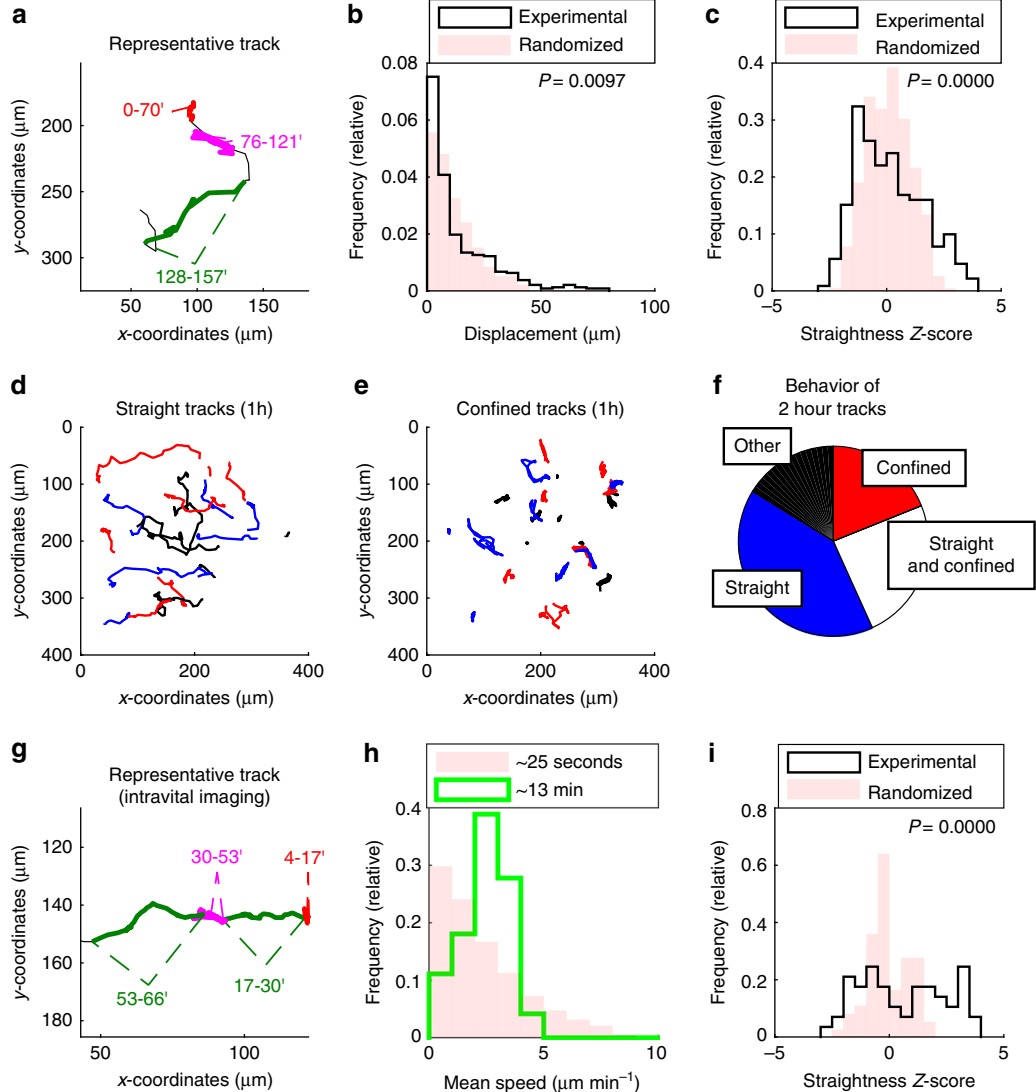

**Fig. 4** Lung-infiltrating T cells switch between straight and confined migration. GFP+ effector T cells were injected i.v. into LPS-treated mice. After two to five days, lungs were explanted (**a–f**) and imaged with two-photon microscopy or imaged using intravital microscopy (**g–i**). **a** Representative track displaying straight (*green*) and confined (*red* and *magenta*) migration. Numbers indicate time in minutes. **b, c** Analysis of displacement (**b**) and straightness Z-score (**c**) of 15 min tracks (n = 477 tracks). P-values were computed with the Kolmogorov–Smirnov test. **d, e** Analysis of 1 h tracks (n = 94 tracks). Depiction of "straight" (**d**; straightness Z-score > 1.7) and "confined" tracks (**e**; straightness Z-score < −0.9). **f** Two hour tracks (n = 37) were analyzed for switching between confinement and straight migration. Quantitative analyses from **a–f** were performed on pooled data obtained from three independent experiments. **g–i** GFP+ effector T cells were injected i.v. into LPS-treated mice. After two to five days, lungs were imaged with intravital two-photon microscopy. **g** Representative track displaying straight (*green*) and confined (*red* and *magenta*) migration. Numbers indicate time in minutes. **h** Shown are histograms of the mean speeds of 13 min (n = 154 tracks; median speed: 2.4 μm min⁻¹) and 25 s (n = 5194 tracks) tracks. **i** Analysis of straightness Z-score (track duration approximately 13 min) before and after randomization of velocity vectors (n = 154 tracks). P-value was computed with the Kolmogorov–Smirnov test. A total of three movies obtained from different mice were analyzed in **g–i**

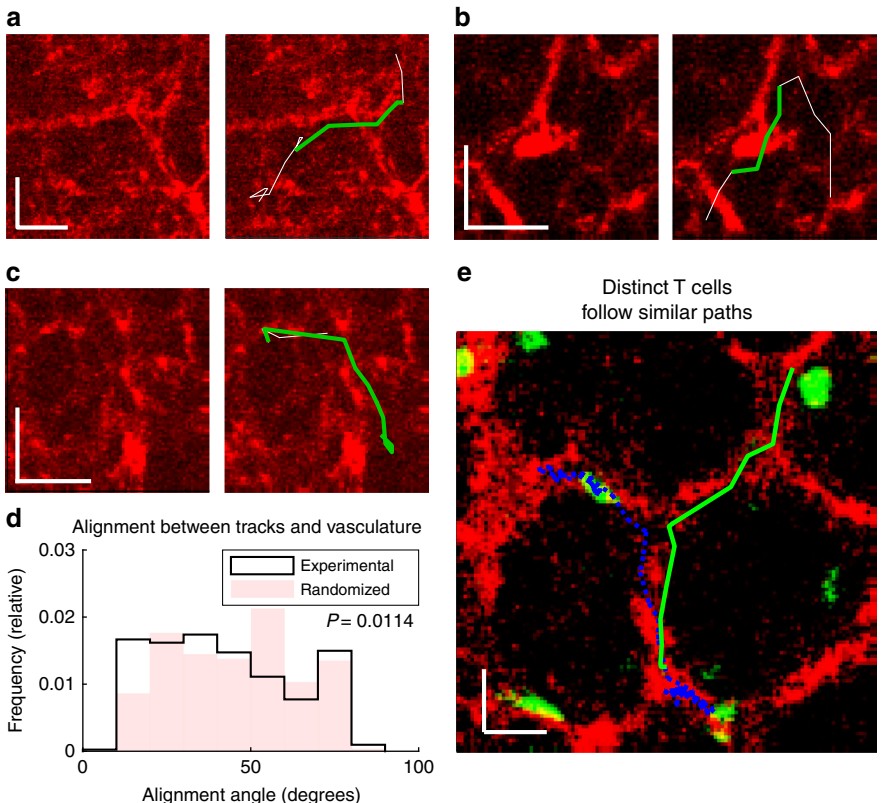

**Fig. 5** Guidance of lung-infiltrating T cells along the vasculature. GFP[+] effector T cells were injected i.v. into LPS-treated mice. After two to five days, the vasculature was labeled intravitally by intravenous injection of fluorescently labeled lectin. Two-photon imaging of explanted lungs enabled simultaneous visualization of GFP[+] T cells and the vasculature. **a–c** High-resolution images of vascular stain of live lung tissue (*left panels*) with superimposed T cell trajectories (*right panels*). Tracks that are apparently aligned with the vasculature are highlighted in green. *Scale bars* = 20 μm. **d** Tracks were split into 24 μm wide segments (*n* = 414 segments from ten movies). For each experimental segment, a randomized control segment was generated. An alignment angle, between segments and the main orientation of the vasculature, was calculated for all experimental and randomized segments. The Mann–Whitney test was used to calculated *P*-values for comparison of alignment angles from experimental and randomized segments. **e** High-resolution image of lung region displaying adoptively transferred T cells (in *green*) and lung vasculature (in *red*). The *blue dotted line* and the *green solid line* indicate tracks of two distinct T cells. *Scale bar* = 20 μm

Visual inspection revealed that directionally persistent and confined T cells frequently moved along neighboring vessels (Fig. 5a–c, *bold green lines*; Supplementary Movies 5–7). We also observed T cells that crossed vessels or that moved in the "empty" space in between vessels (Fig. 5a, b, *white thin lines*). To objectively quantify alignment between T cell movement and the vasculature, we employed a customized computational approach. Briefly, for each T cell track segment with a maximum displacement of 24 μm, an "alignment angle" was determined, which measures how well the T cell track is aligned with the main orientation of the surrounding vasculature (0 degrees: perfect alignment; 90 degrees: perpendicular crossing of vessel; for a more in depth discussion of methodology, see methods and Supplementary Fig. 3). As a control, we generated for each experimental track a randomized track derivative and measured its alignment angle. We found that experimental tracks were slightly but significantly better aligned with the vasculature than randomized control tracks (Fig. 5d).

In addition, we analyzed multiple distinct T cells that moved consecutively through the same tissue volumes. We found that T cells tended to move along very similar paths, even when they traversed the space at distinct time-periods (Fig. 5e; compare *blue* and *green* tracks; Supplementary Movie 8). Moreover, such tracks were frequently aligned with the surrounding vasculature (Fig. 5e). These results suggested that contact guidance by the vasculature

influences effector T cell movement in the lung upon acute lung injury.

**Chemokines promote speed but not persistence of lung T cells**. Next, we explored whether chemokines contribute to environmental guidance of effector T cell migration within inflamed lung tissue. While recent two-photon imaging studies have shown that in some cases chemokine trails guide tissue-infiltrating T cells[42], the impact of chemokines on interstitial immune cell migration is more complex than originally envisaged[43].

Similar to previous studies, we used pertussis toxin (PTX), an inhibitor of chemokine receptor-dependent Gα$_i$-type G protein signaling, to study effects of chemokines in vivo[11, 38, 44, 45]. Briefly, we captured the migration of lung-infiltrating T cells as described above using two-photon microscopy with our imaging model of acute lung injury ("before PTX"). We then added PTX directly to the superfused medium. After incubating the lungs for 2 h to ensure effective PTX inhibition, we measured T cell migration again ("after PTX"). Migration before and after treatment was measured in identical imaging regions, which excluded a confounding influence due to environmental heterogeneity. Analysis of average track speeds revealed a significant reduction after treatment with PTX, similar to previous observations in lung tumors and infected brain[11, 38] (median

speeds: before treatment: 2.4 μm min$^{-1}$; after treatment: 2.0 μm min$^{-1}$; Fig. 6a; Supplementary Movie 9). Consistent with the reduction in speed, time vs. mean squared displacement measurements revealed that PTX-treatment also reduced rate of tissue infiltration by lung-infiltrating T cells by 50% (Fig. 6b; motility coefficients: before treatment: 6.80 μm$^2$ min$^{-1}$; after treatment: 3.59 μm$^2$ min$^{-1}$). Interestingly, when we analyzed T cell tracks for directional persistence using the straightness Z-score as previously described, PTX-treatment did not lead to significant changes (Fig. 6c). This suggested that while chemokines deposited within lung tissues can regulate T cell speed, intra-tissue chemokine gradients were dispensable for the intermittent migration of lung-infiltrating T cells. This was corroborated by in vitro experiments, where PTX inhibited spontaneous T cell migration through transwells even when no external chemokine gradients were generated (Supplementary Fig. 4a). Moreover, chemokines CXCL10, CCL21, and CXCL12 all induced migration (Supplementary Fig. 4b).

**ROCK is required for tissue-navigation of lung T cells.** RhoA-ROCK-myosin II pathway has been shown to be important in promoting the speed of naive T cells navigating within lymph nodes[14, 15, 17]. We also found that in vitro, chemokines can induce ERK and RhoA-ROCK signaling pathways in CTL (Supplementary Fig. 4c, d), as shown previously for other cell types[17, 46–48]. We then tested whether ROCK promotes migration

of effector T cells within the inflamed lung environment. To this end, we used our two-photon imaging model of acute lung injury and tracked lung-infiltrating T cells before and after addition of the ROCK inhibitor Y-27632. The ROCK inhibitor strongly reduced the average track speed of lung-infiltrating T cells (median speeds: before treatment: 2.5 μm min$^{-1}$; after treatment: 0.8 μm min$^{-1}$; Fig. 7a; Supplementary Movie 10). ROCK inhibition also led to a massive reduction of the motility coefficient, as determined by time vs. MSD analysis (Fig. 7b; motility coefficient: before treatment: 7.08 μm$^2$ min$^{-1}$; after treatment: 1.65 μm$^2$ min$^{-1}$). Importantly, the ROCK inhibitor also led to a significant reduction of the straightness of analyzed tracks (Fig. 7c). Notably, inhibition of ROCK impaired migration of lung-infiltrating T cells more strongly than inhibition of chemokine signaling (compare Fig. 6 with Fig. 7), suggesting that ROCK-activity is only partially dependent on chemokine-dependent Gα$_i$ signaling. This was confirmed in vitro, where treatment with the ROCK-inhibitor Y-27632 reduced spontaneous transwell migration much more strongly than PTX (Supplementary Fig. 4a). These results showed that ROCK-dependent cytoskeletal remodeling is a major requirement for switching between fast and directionally persistent migration of lung-infiltrating T cells during acute lung injury.

**Intermittent migration increases T cell contacts with target.** To determine the potential role of intermittent migration as a

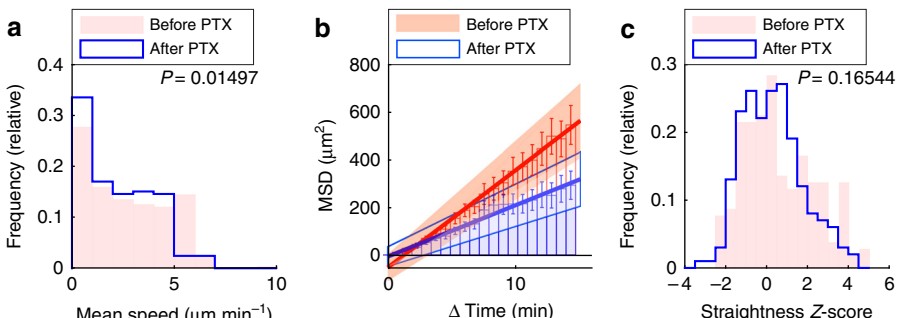

**Fig. 6** Chemokines promote speed but not directionally persistent T cell migration. GFP$^+$ effector T cells were injected i.v. into LPS-treated mice. After two to five days, lungs were explanted and imaged with two-photon microscopy. Identical regions were imaged before and after addition of pertussis toxin (PTX). Tracks with a duration of 15 min were analyzed (n = 205 tracks both before and after treatment). **a–c** Mean speed (**a**), time vs. mean squared displacement (MSD) (**b**), and straightness Z-score (**c**) before and after treatment with inhibitor. (**b**) Bars show mean squared displacement. Also shown are standard errors (error bars), linear regression (line) and confidence intervals (shaded areas). Shown are pooled data from experiments with four distinct mice (**a**, **c**) or data from a single representative experiment (**b**). P-values were computed with the Mann–Whitney test

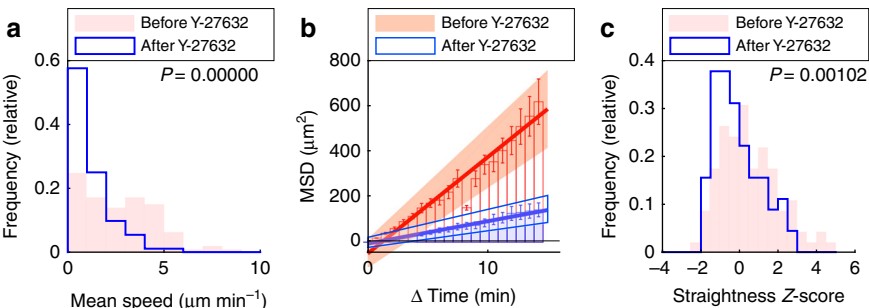

**Fig. 7** ROCK is required for straight lung T cell migration. GFP$^+$ CD8$^+$ effector T cells were injected i.v. into LPS-treated mice. After two to five days, lungs were explanted and imaged with two-photon microscopy. Identical regions were imaged before and after addition of Y-27632. Tracks with a duration of 15 min were analyzed (n = 92 tracks both before and after treatment). **a–c** Mean speed (**a**), time vs. mean squared displacement (MSD) (**b**), and straightness Z-score (**c**) before and after treatment with inhibitor. **b** Bars show mean squared displacement. Also shown are standard errors (error bars), linear regression (line), and confidence intervals (shaded areas). Shown are pooled data from experiments with three distinct mice (**a**, **c**) or data from a single representative experiment (**b**). P-values were computed with the Mann–Whitney test

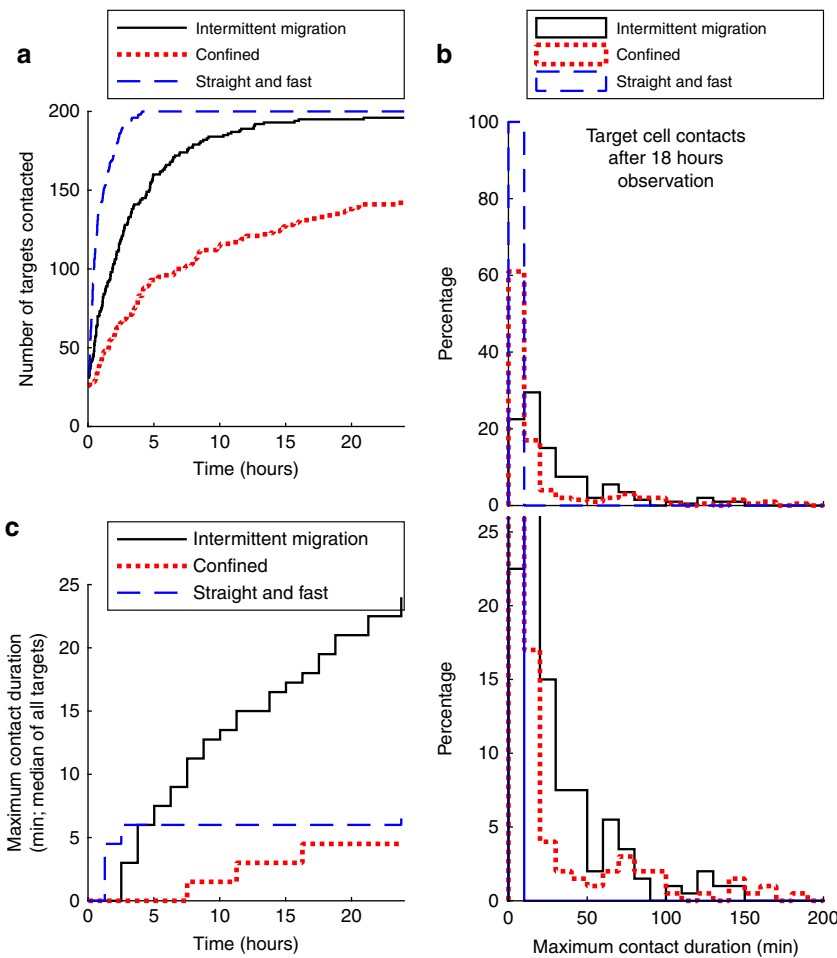

**Fig. 8** Intermittent migration increases long-lasting T cell target cell contacts. Contacts between 200 motile T cells and 200 target cells within a 1 mm² region were quantified using a computational model (see Methods). **a** The number of target cells that were contacted by any of the T cells is shown over time. **b** Eighteen hours after the initiation of the simulation, for each target cell the longest continuous contact with a T cell was calculated. The percentage of different contact durations is shown for each migration mode. *Upper panels* and *lower panels* have a different range on the y-axis, so that the percentage of short (*upper panel*) and long (*lower panel*) contacts can be seen. **c** The contact duration between T cells and targets was calculated as described in panel **b**, for various chosen observation periods as indicated on the x-axis. For each datapoint, the median of all contact durations is shown

strategy for T cell immunity in inflamed lung, we used computational modeling to compare the interaction dynamics of T cells that display purely confined or straight migration with those that display an intermittent migration pattern (see Methods and Supplementary Movies 11 and 12). Briefly, we studied an area of 1 mm² populated by 200 T cells and 200 target cells and modeled migration and T cell contacts for 25 h. This analysis revealed that T cells that moved straight with a constant speed of 6 µm min⁻¹ rapidly engaged in brief contacts with the majority of target cells (50% were contacted after half an hour; Fig. 8a). T cells that showed intermittent migration showed intermediate efficiency (50% of targets contacted after 2 h; Fig. 8a) and confined T cells

showed the poorest capacity to engage in brief contacts with target cells (50% contacted after 6 h; Fig. 8a).

A different picture emerged when we assessed the duration of interactions, which is relevant since long-lasting contacts by CTLs might be an important prerequisite for the induction of lytic cell death of targets[49]. Analysis of the maximum continuous contact of each target cell with a T cell revealed that in the straight T cell group only short contacts were established (Fig. 8b; *blue dashed line*). In the confined group, many targets established no contacts at all, also leading to short contact durations (Fig. 8b; *red dotted line*). In contrast, intermittent migration led to an enrichment of longer-lasting contacts between 10 and 100 min (Fig. 8b; *black*

*solid line*). Time-course analysis for different observation periods confirmed this result. Median contact duration remained below 6 min for straight and confined T cells, whereas the contact duration of intermittent T cells increased continuously (Fig. 8c). These results demonstrated that intermittent T cell migration can allow target identification in space while also enhancing the duration of continuous contact times between T cells and target cells.

## Discussion

In order to provide protection against infections and cancer, T cells must re-circulate within organisms and migrate through individual tissues. In contrast to other organs, such as the skin and the gut, the molecular mechanisms that regulate T cell migration in the lung have remained relatively poorly characterized[50]. In the present study, we provide novel insights into the molecular mechanisms that regulate T cell motility within the interstitial space of inflamed lungs using a mouse model of acute lung injury. We find that T cells display an intermittent migration pattern, characterized by transitions between confinement and straight relocation. The directionality of T cells is not random but partially aligned with the architecture of the lung tissue. Mechanistically, the RhoA-ROCK-myosin II signaling pathway enabled high-speed straight migration of lung-infiltrating T cells. Computational studies provided proof-of-principle that intermittent migration balances the need for spatial coverage as well as the establishment of long-lasting contacts between T cells and target cells.

We use two-photon imaging to obtain a direct view of T cell behavior at high spatial and temporal resolution within live lung tissue. Importantly, we track T cells for up to 3 h, whereas in many previous studies the analyzed track duration was usually in the range of 10 to 40 min[11, 28–31]. The generation of long continuous trajectories is feasible because we captured movie sequences of several hours duration and performed correction for tissue-drift, a common problem during two-photon imaging[51]. Furthermore, in our system the T cells move with relatively low speeds, so that only very few cells "escape" from the image volume during movie capture. The generation of such long observation periods is crucial because shorter timescales gave the misleading impression that the T cell population is heterogeneous and composed of a subset of cells that do not move at all while other cells move with high speed and directional persistence (Fig. 2). Longer movies suggest that in fact individual T cells switched between periods of confinement, lasting typically tens of minutes up to hours, and ballistic relocation.

We analyze two-photon imaging with customized statistical tests to verify the visual impression that lung-infiltrating T cells indeed switch between distinct migration modes seen in both explanted and intravital lung imaging. Such intermittent migration patterns have been observed in a variety of organisms from protozoa to mammals[52], but their role for T cell migration in live tissues has remained unclear. For example, Fourier analysis has either supported or argued against periodic transitions between active migration and pausing of naive T cells in lymph nodes[37, 53]. In another study, intermittent T cell migration in infected brain was supported by good fits between empirical data and computer simulations, but direct observation of intermittent migration was lacking[38]. In the present study, comparison of empirically determined T cell trajectories with randomized tracks and theoretical models strongly supports the conclusion that lung-infiltrating T cells indeed display intermittent confinement and straight ballistic motion that results in long-distance migration.

Theoretical computational models have shown that, under appropriate conditions, intermittent migration can minimize search time for targets of unknown location[54]. On the other hand,

intermittent migration of T cells could be an attempt to balance thoroughness and extensiveness of search for target cells[3, 36], similar to a previously described migration model where T cell meandering increased continuous contacts with target cells[2]. Our computational model suggests that intermittent migration includes both periods of ballistic relocation leading to increased target identification while periodic confinement promotes more intimate and longer-lasting contacts with target cells. This could be important because some estimates have suggested that full T cell activation may require up to 10 min of contact, depending on peptide affinity and amount[55]. Moreover, CTLs are relatively inefficient at killing individual target cells in vivo, and cumulative contacts of 10–60 min have been shown to occur before target cell lysis[49]. Intermittent migration may balance the need to search a large area such as inflamed lung tissue while also allowing for long lasting contacts leading to CD8 T cell activation and lytic function.

Our study also provides novel insights into the molecular mechanisms that regulate T cell migration within inflamed lung parenchyma. While several recent studies have reported imaging of T cell migration within live lung tissue, the mechanisms that regulate migration of T cells infiltrating the lung parenchyma have remained essentially uninvestigated[28–31]. Chemokines are well-established promoters of recruitment of T cells to the lung and other tissues[56]. In contrast, the mechanisms that influence chemokine-dependent migration of T cells within the interstitial space of diverse tissues are more complex and cannot simply be explained by motility along pre-formed chemokine gradients. For example, while track analysis of T cells in the skin was consistent with chemotaxis[4, 5], the bias was very subtle and only detectable with sophisticated analytical tools[5]. In the brain, inhibition of chemokine receptor-dependent $G\alpha_i$-type G protein signaling with PTX reduced speed of CTL by <50% but did not change motility qualitatively[38]. Similarly, PTX reduced the motility coefficient of T cells in lung tumors by 50%[11]. Here, we report that treatment with PTX has a similar effect on the motility of CTL within inflamed lungs: motility coefficients went down by 50%, but only a marginal effect on the intermittent mode of migration was observed as PTX-inhibited T cells still showed both confined and directionally persistent migration. While we still do not know a precise mechanism for ROCK activation by chemokines in vivo, our results suggest that chemokines are not required for the straight migration of T cells in the lungs, which is distinct to guidance by chemokine trails in the trachea of influenza-infected animals[42].

Motility of effector T cells and other leukocytes within dense tissues likely require dramatic cytoskeletal reorganization. We targeted the RhoA-ROCK-myosin II pathway, which is specifically involved in leukocyte squeezing motion, particularly in integrin-independent T cell movement[14–16]. Indeed, inhibition of ROCK led to a massive reduction of both speed and directional persistence of lung-infiltrating T cells. It is striking that inhibition of the ROCK-pathway leads to a more pronounced reduction of migration than inhibition of chemokine-dependent $G\alpha_i$ signaling. This may indicate that chemokines activate ROCK at least partially via $G\alpha_i$-independent pathways[46], or that activated T cells have functional levels of active ROCK independent of chemokine-induced signaling.

Our study also provides novel insights with regard to guidance by environmental structures. It is already known that CD4[+] effector T cells in the skin move along fibers comprised of extracellular matrix in an integrin-dependent manner[10]. Naive T cells move along a fibroblastic reticular cell (FRC) network in lymph nodes, which is presumably mediated by deposited chemokines[57]. Recent reports also suggest that FRC-like networks mediate migration of effector T cells in infected brain tissue[12, 58].

Our data extend these findings and corroborate that tissue-specific structures within individual tissues are important in mediating the type of motility taken by T cells. Altogether with the importance of the ROCK pathway, these data support the hypothesis that lung-infiltrating T cells employ a squeezing type of migration along a "path of least resistance", a previously proposed model of leukocyte migration in live tissues[13, 16, 53].

Our findings may provide a rationale for novel treatments of inflammatory lung diseases by targeting T cell motion in inflamed lung. Indeed, targeting molecules that are involved in T cell trafficking and migration, such as integrins and chemokines, is the basis of established and emerging treatments of inflammatory diseases, including psoriasis, multiple sclerosis, Crohn's disease and asthma[50]. In the present study, we show that the ROCK pathway plays a crucial role for T cell migration within inflamed lung, extending previous observations that this pathway promotes migration of naive T cells in the lymph node[17]. This suggests that localized targeting of the ROCK pathway may be useful for the treatment of acute lung injury and other inflammatory diseases that depend on T cell navigation of inflamed lung tissue mediating inflammatory damage. In fact, targeting of the ROCK pathway is an emerging treatment option for glaucoma[59]. In conclusion, our study sheds new light on effector T cell function in an inflammatory tissue: quantitative analysis coupled with inhibition of specific molecular pathways reveals novel insight into the role of chemokines and ROCK in mediating effector T cell migration in inflamed lung.

## Methods

**Mice and reagents**. All experiments were performed with C57BL/6 (Jackson Laboratories) and B6.Ubiquitin-GFP mice[60] (Jackson Laboratories). Typically, female animals were used between 8–20 weeks of age. Breeding and maintenance of animals used in this research conform to the principles outlined by the Animal Welfare Act of the National Institutes of Health. The protocol for animal work was approved by the IACUC at the University of New Mexico (protocol # 15-200328-HSC). All efforts were made to minimize suffering with use of ketamine and xylazine when appropriate. Euthanasia was performed by isoflurane overdose. Anti-CD3 (145-2C11, BioXCell) and anti-CD28 (PV-1, BioXCell) antibodies were used for T cell priming. For intravascular T cell staining, anti-CD3-PerCPCy5.5 (145-2C11, eBioscience; flow experiments) and anti-CD3-Alexa 647 (17A2, BDPharmingen; confocal imaging) were used. For intranasal staining, an anti-CD45-PE antibody (30-F11; Molecular Probes) was used. In inhibitor experiments, the ROCK inhibitor Y-27632 (Calbiochem) was used at a final concentration of 20 μM. PTX (Sigma) was used at a final concentration of 100 ng ml$^{-1}$. Anti-CD8-APC-Cy7 (53-6.7, eBioscience) was used for staining single cell suspensions. For western blotting, anti-p-ERK (Thr202/Tyr204, Cell Signaling Technology #9101, used at 1:1000), anti-ERK (137F5, Cell Signaling Technology, used at 1:1000) and anti-actin antibodies (AC-15, Sigma, used at 1:5000) were used.

**Generation of CD8$^+$ effector T cells**. Splenocytes from ubiquitin-GFP mice were primed with anti-CD3 (0.5 μg ml$^{-1}$) and anti-CD28 (0.5 μg ml$^{-1}$) antibodies, followed by stimulation with 20 U ml$^{-1}$ IL-2 (National Institutes of Health) every other day. Six to eight days after priming, the resulting cell population (~ 90% CD8$^+$ T cells) was used for adoptive transfer experiments.

**Experimental model of acute lung injury**. Mice were anesthetized intraperitoneally with 90 μg ketamine and 8.1 μg xylazine per gram body mass and received 150 μg LPS intranasally (Escherichia coli K-235, Sigma). After one or two days, $10 \times 10^6$ GFP$^+$ effector T cells were injected via the tail vein. Two to five days after LPS injection, the lungs of such "LPS-treated mice" contained inflammatory foci with GFP$^+$CD8$^+$ effector T cell infiltrates.

**Flow cytometry analysis**. Flow cytometry analysis was performed to quantify the percentage of intra-airway and/or intravascular T cells. Staining of intra-airway T cells was achieved by anesthetizing mice with ketamine/xylazine, followed by i.n. injection of 0.25 μg anti-CD45 antibody (30-F11; Molecular Probes; used at 1:100). For intravascular staining, mice received 3 μg of anti-CD3 antibody via the tail vein, followed after 5 min by euthanasia. Single-cell suspensions were generated according to standard immunology protocols (lymph nodes) or with a commercial tissue-dissociation kit (lung; Miltenyi, 130-095-927), followed by staining with an anti-CD8 (53-6.7, eBioscience; used at 1:100) antibody and analysis with an LSR Fortessa (BD). Gating strategies are shown in Supplementary Fig. 5.

**Confocal imaging of fixed lung tissue**. LPS-treated mice received 3 μg of anti-CD3-Alexa 647 (17A2, BDPharmingen) antibody via the tail vein, followed after 5 min by euthanasia. Lungs were inflated with 2% agarose via the trachea[29], and fixed with 4% paraformaldehyde. The convex surface of such lungs was imaged with a ZEISS LSM510 META/NLO confocal microscope. GFP and Alexa 647 were excited simultaneously with Argon (488 nm) and HeNe (633 nm) lasers. Emitted fluorescent light was detected with photomultiplier tubes (PMT) after separation with a suitable filter cube (dichroic mirror: 545 nm; bandpass: 500–530 nm; bandpass: 650–710 nm).

**Two-photon imaging set-up**. Two-photon imaging was performed either with an upright Bio-Rad microscope or an inverted ZEISS LSM510 META/NLO (capturing equivalent T cell behavior). Briefly, with the Bio-Rad setup, GFP and DyLight594 were excited simultaneously with a laser tuned to a wavelength of 800 nm and detected with PMTs after separation of the emitted fluorescent light with a suitable filter cube (dichroic mirror: 575 nm; bandpass filters: 510/40 nm and 670/45 nm). Movie sequences were captured with the LaserSharp2000 software. For the ZEISS LSM510 META/NLO imaging set-up, a Chameleon Ti:Sapphire laser tuned to 800 nm (Coherent) and suitable filter cube (dichroic mirror: 560 nm; bandpass filters: 500–550 nm and 575–640 nm) were used for specific detection of GFP and DyLight594. Movies were captured with the ZEN user interface (Zeiss). In both systems, time-lapse movies were composed of a series of image volumes (depth: 84 to 156 μm; step size: 6 or 12 μm) separated by 90" (movie duration >2 h) or 45" (movie duration < 2 h).

**Lung preparation for live imaging**. The lung vasculature of LPS-treated mice was labeled intravenously (i.v.) with 70 μg DyLight594-labeled lectin (from Lycopersicon esculentum, Vector Laboratories). After euthanasia, the lungs were surgically dissected and inflated intratracheally with low gelling temperature agarose (2% dissolved in PBS, Sigma)[29]. After transfer into a customized imaging chamber (for imaging with the Bio-Rad microscope) or a Chamlide AC-B25 imaging chamber (Live Cell Instruments; for the ZEISS microscope), mechanically stabilized lungs were superfused with oxygenated DMEM (Dulbecco's Modified Eagle's medium) (Gibco, 21063-045; 5% serum, Atlanta Biologicals; 100 units ml$^{-1}$ penicillin and 100 μg ml$^{-1}$ streptomycin, Gibco) and maintained continuously at 37 °C.

**Intravital two-photon imaging**. Mouse lung intravital two-photon imaging was performed according to published procedures[61] with the following modifications to eliminate the use of ketamine/xylazine and improve tissue stability during imaging. Mice were anesthetized with 310 mg kg$^{-1}$ of Avertin given i.p. and intubated orotracheally with a 20G angiocatheter. Following intubation, mice were ventilated using a MiniVent Ventilator (Model 845, Harvard apparatus) with oxygen and 1.5% isoflurane at a rate of 200 breaths per minute and with a tidal volume of 10 μl per gram body weight (typically 200 μl for a 20 g mouse). If the mouse showed any signs of spontaneous breathing it was given a supplemental ½ dose of Avertin before surgery. The lung was exposed through a left thoracotomy and the dorsal surface of the left lobe (~ 1/3 of the lobe) was attached to a thin flexible plastic support using a thin film of VetBond. The mouse is placed on the lower plate of a custom built chamber and the upper plate of the chamber is assembled and lowered over the exposed left lung. The plastic support attached to the bottom of the lung is carefully lifted to bring the lung into contact with the cover glass lining the bottom of the upper chamber plate and the plastic support secured with a hair-pin. The mouse is maintained at 37 °C by warming both upper and lower plates using a Warner TC-344C Dual Channel Temperature Controller (Warner Instruments). Anesthesia is maintained with 1.5–2% isoflurane during imaging. Mice are imaged for up to 4 h and given supplemental fluids (150 μl saline s.c.) for any experiment lasting more than 1.5 h. The imaging procedure is terminal and at the end of the experiment anesthesia is increased to 3% isoflurane and mice are euthanized by cervical dislocation while deeply anesthetized. The upper chamber is filled with water and video-rate and time-lapse imaging were performed with a custom built two-photon microscope equipped with a 1.0 NA 20× water dipping objective (Olympus) running ImageWarp acquisition software (A&B Software) as previously described[61, 62]. Tissue perfusion was verified after the imaging preparation was completed by evaluating superficial pulmonary capillaries under bright field and by injecting 655 nm Q-dots (ThermoFisher) and performing video-rate imaging to confirm robust pulmonary vessel flow. Chameleon Vision II Ti:Sapphire laser (Coherent) tuned to 910 nm was used to excite fluorescence and emission detected by PMTs using 495 nm and 560 nm dichroic filters: Blue (<495 nm, SHG collagen), green (495–560 nm, eGFP), and red (>560 nm, 655 Q-dots). Auto fluorescence of alveolar macrophages appears as mix of color (495–600 nm) and thus can be discriminated from the eGFP signal. For time-lapse imaging, we acquired a 300 × 340 × 75 μm volume as 31 sequential 2.5 μm z-steps with a time resolution of approximately 25 s. For video-rate imaging, we acquired a 300 × 340 μm field at ~ 40 ms frame$^{-1}$ time resolution for 500 continuous frames. X-Y resolution was 0.75 μm pixel$^{-1}$, which resolved individual cells and small capillaries. Multi-dimensional data sets were analyzed with MatLab.

**Image-sequence processing and cell tracking**. Image-analysis was performed with MatLab, after importing ".pic" (Bio-Rad), ".lsm" (Zeiss) or ".stk" files

(intravital imaging). Tissue drift was corrected by aligning consecutive image-volumes based on the intensity pattern of the lectin stain. Cell-tracking was performed manually by identification of the centroids of individual cells at consecutive timepoints. When multiple cells collided, tracking was continued if individual cells maintained a consistent forward momentum. Tracking was performed in three dimensions. Nevertheless, because of the lower resolution in the Z-dimension, several analyses, including speed autocorrelation, turning angle autocorrelation, curve fitting, randomized track parameters and environment alignment were performed on coordinate projections on the XY-plane.

**Curve fitting**. Curve fitting was performed similar to a previously published approach[36]. Briefly, we used maximum likelihood estimation to parameterize candidate probability distribution functions (PDF). Probability model parameters were fitted using CDF, which is advantageous over the usage of binned data[36]. In the present paper, we fit Rayleigh, power law and log-normal distributions. The Rayleigh distribution was fit with the MatLab function *raylfit*. Fits for power law and log-normal distributions were performed as described previously[36]. The goodness of fit of each candidate PDF to empirical data was evaluated using corrected Akaiki Information Criterion (AICc), negative log-likelihood measures and the Kolmogorov–Smirnov (KS) test.

**Autocorrelation of speed**. To quantify whether individual T cells switched between different speed states, i.e., whether over time similar speeds clustered together, we calculated for each individual track an autocorrelation function for speeds. Briefly, we computed the speed of each frame-to-frame velocity vector **v**. For each single track, we then collected all speed pairs $sp$ that are separated by defined time intervals $t$. By computing the Spearman correlation coefficient for all $n$ speed pairs $sp$ of time intervals $t$ with the MatLab function *corr*, we obtained a speed autocorrelation function for each track:

$$C_{\text{track}}(t) = corr([sp_1; sp_2; sp_n],' \text{Spearman}').$$ (1)

By averaging the track autocorrelation function over all $t$ tracks, we obtained a speed autocorrelation function for the entire population:

$$C_{\text{population}}(t) = \frac{1}{t} \sum_{x=1}^{t} C_{\text{track}}(t).$$ (2)

**Time vs. mean squared displacement**. As a quantitative measure for the rate of tissue-infiltration, we measured the MSD over time. Briefly, we calculated, for each individual track, the three-dimensional displacement $x$ of cell positions separated by defined time intervals $t$. For each time interval $n$ different time delays were included when the corresponding measurements were not overlapping temporally. We calculated, for each track, the MSD of $n$ displacements of a given time interval:

$$\text{MSD}_{\text{track}}(t) = \frac{1}{t} \sum_{x=1}^{n} x_i(t).$$ (3)

By averaging the MSD of all $t$ tracks, we obtained the MSD for the entire population:

$$\text{MSD}_{\text{population}}(t) = \frac{1}{t} \sum_{x=1}^{t} \text{MSD}_{\text{track}_i}(t).$$ (4)

We used the MatLab function *regress* to perform linear regression, which yielded the slope of the linear fit and its 95% confidence interval. Based on linear regression we computed the motility coefficient $M$ in dimension $D = 3$ of analyzed cell populations:

$$M = \frac{\text{Slope}_{\text{Linear Regression}}}{2xD} \mu m^2 min^{-1}.$$ (5)

**Autocorrelation of turning angles**. To quantify directional alignment within individual tracks over time, we computed the turning angle autocorrelation, similar to a previously described approach[63]. Briefly, we collected, for each single track, all vector pairs separated by defined time intervals $t$. We then measured the cosine of the turning angle $\theta$ of all vector pairs. The cosine has a value of minus one for vectors that point in opposite directions and one for vectors that point in same direction. By calculating the mean of all $n$ cosines per time interval, we obtained an autocorrelation function for individual tracks:

$$C_{\text{track}}(t) = \frac{1}{n} \sum_{i=1}^{n} \cos(\theta_i(t)).$$ (6)

By averaging the track autocorrelation function over all $t$ tracks, we obtained an autocorrelation function for the entire population:

$$C_{\text{population}}(t) = \frac{1}{t} \sum_{x=1}^{t} C_{\text{track}}(t).$$ (7)

**Computation of the straightness Z-score**. For each experimental track $t_{\text{exp}}$, we generated 100 randomized track-derivatives $t_{\text{rand}_i}$. The velocity vectors in the randomized tracks had identical magnitudes as in the experimental track, but their orientation and order were randomized (Supplementary Fig. 1a–d). We then calculated the displacement between the start and end of the experimental track $d_{\text{exp}}$, as well as the displacements of the 100 randomized tracks $d_{\text{rand}_i}$. The experimental straightness Z-score $Z_{\text{exp}}$ shows how many standard deviations $\sigma$ the experimental displacement deviates from the expected displacement $\mu$ of all randomized track derivatives (Supplementary Fig. 1e):

$$Z_{\text{exp}} = \frac{d_{\text{exp}} - \mu(d_{\text{rand}_i})}{\sigma(d_{\text{rand}_i})}.$$ (8)

Negative Z-scores suggest that the experimental displacement is less than expected by chance, and positive Z-scores suggest that the experimental displacement is higher than expected by chance. For control purposes, we also calculated a "randomized" straightness Z-score $Z_{\text{rand}}$, which shows how many standard deviations $\sigma$ one single randomized displacement deviates from the expected displacement $\mu$ of all randomized track derivatives:

$$Z_{\text{rand}} = \frac{d_{\text{rand}_1} - \mu(d_{\text{rand}_i})}{\sigma(d_{\text{rand}_i})}.$$ (9)

**Switching between confinement and straight migration**. The straightness index of 15 min track children within 2 h parent-tracks was measured. Parent tracks were categorized as straight when >10% of track children had a straightness index >1.7. Parent tracks were categorized as confined when >7% of track children had a straightness index <−1.7.

**Analysis of alignment between trajectories and vasculature**. To quantify the alignment of T cell trajectories with the surrounding vasculature, we analyzed image-sequences containing cell-trajectories and a channel with a lectin-stain of the vasculature. To simplify the analysis, each track was split into track-segments so that each segment had a maximum displacement of roughly 24 μm. We then processed each segment with an image analysis algorithm and obtained an "alignment angle" that measures how much the segment deviates from the main orientation of the vasculature, with a range from 0 degrees (perfect alignment) to 90 degrees (perpendicular movement over vessel).

Briefly, each segment was converted from a series of vectors into an "original" rasterized pixel-image (Supplementary Fig. 3a, b). The rasterization enabled a pixel-by-pixel comparison between the track-segment and the corresponding intensity of the vascular stain (Supplementary Fig. 3d). Specifically, we calculated the median "overlap intensity" of all vascular pixels that overlapped with the given segment, and we similarly measured the overlap intensity of a perpendicular segment (Supplementary Fig. 3d). If the analyzed segment is well aligned with the vasculature, we expect a high value of the "overlap ratio" between the overlap indices of given and perpendicular segments.

Computation of the overlap ratio enabled quantification of how much the original segment deviated from the main orientation of the vasculature. Briefly, we generated rotated and/or translated segment "derivatives" of the original segment image, calculated the overlap ratio for each segment derivative (Supplementary Fig. 3c, e, f) and then isolated the 5% segment derivatives with the highest overlap ratio. These segments are best aligned with the vasculature and, therefore, their orientations provide an indication of the main orientation of the vasculature. We thus calculated how much the original segment deviates from each of the best aligned segment derivatives ("rotation angle": 0 to 90 degrees; Supplementary Fig. 3g, h). The median of all rotation angles was defined as the alignment angle between the original segment and the main orientation of the surrounding vasculature (Supplementary Fig. 3h).

**Computational modeling of T and target cell interactions**. To quantify the contact duration between motile T cells and target cells, we simulated T cell migration in an area of 1 mm$^2$ for 25 h. Initially, the region was populated randomly with 200 T cells of $10 \times 10$ μm$^2$ size (no overlap between different T cells was

allowed) and analogously with 200 target cells. Target cells were stationary. For each T cell, a trajectory was created that contained 1000 positions, separated by time-steps of 90 s duration. The specifics of how these trajectories were generated depended on the type of migration that was simulated, i.e., straight, confined, and intermittent. Straight T cells moved with a constant speed of 6 µm min$^{-1}$ and obtained a random orientation that they maintained throughout the simulation. The trajectories of the confined cells were based on experimentally obtained T cell tracks. Briefly, all tracks were split into subtracks with a duration of 20 timesteps and these subtracks were filtered so that only confined subtracks (straightness Z-score < 1.5) remained. For each simulated T cell, these tracks were stitched together randomly so that tracks of sufficient duration (>=1000 timesteps) were obtained. Tracks for intermittent migration were obtained directly from experimental data. For each simulated T cell, experimental tracks of 2 h duration were stitched together randomly. When a T cell moved out of the field of view, it was randomly re-positioned within the field of view. Contacts between T cells and target cells were registered whenever T cells and target cells shared overlapping pixels.

**Transwell assay.** $5 \times 10^5$ T cells were seeded on top of 6.5 mm polycarbonate membrane transwell inserts with a pore size of 5 µm or on polyester inserts with a pore size of 3 µm (Corning). 0.5 µg ml$^{-1}$ chemokines were added to the bottom of each well. The percentage of cells that moved to the bottom well after 1.25 h was measured by manual counting.

**Western blot analysis.** CD8$^+$ effector T cells were lysed in cold lysis buffer containing 1% triton and phosphatase inhibitors[64]. Protein concentrations were measured with a BCA assay (Pierce, Thermo Scientific). We loaded 40 µg protein lysates per lane and separated them on 12% Tris-glycine gels. After blotting, the membranes were probed with primary and fluorescently labeled LI-COR secondary antibodies (LiCor IRDye 680 conjugated goat polyclonal anti-mouse IgG, highly cross adsorbed for anti-actin AC15; LiCor IRDye 800CW conjugated goat polyclonal anti-rabbit IgG, highly cross adsorbed for anti-pERK and anti-ERK). All secondaries were used at 1:10,000 dilution. The stained membranes were imaged with a LI-COR Odyssey system (Lincoln, NE).

**RhoA activation assay.** Active RhoA was measured similarly to a previous report[65]. Briefly, active GTP-bound RhoA in lysates from activated mouse CD8$^+$ T cells was analyzed in microplates using the RhoA activation G-LISA kit (Cytoskeleton Inc., Denver, CO). Absorbance of microplate wells at 490 nm was read with an Infinite 200 spectrophotometer (Tecan US, Inc. Morrisville, NC).

**Statistical analysis.** For comparison of whether different distributions are distinct, the Kolmogorov–Smirnov test was used (executed with MatLab's *kstest2*). For comparisons of the median of two distinct populations, a two-tailed Mann–Whitney test was used (executed with MatLab's *ranksum* function). Detailed information about statistical tests, including exact *n* values and the number of replicates are provided in the figure legends.

**Data availability.** The data that support the findings of this study are available from the corresponding author upon request.

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

## Acknowledgements

This work was supported by funding from the following: DARPA P- 1070-113237 (MEM), NIH 1R01AI097202 (JLC), the Spatiotemporal Modeling Center (P50 GM085273), the Center for Evolution and Theoretical Immunology 5P20GM103452 (J.L.C.), a James S. McDonnell Foundation grant for the study of Complex Systems (M.E.M.), and by R01 AI077600, U01 AI095550 and the Mucosal Immunity Studies Two-Photon Imaging Web Resources and Training Fund MIST U01 (M.J.M.). Thanks to the UNM Cancer Center Fluorescence Microscopy Facility (P30-CA118100) as well as the BRaIN Imaging Center (P30GM103400) for help with two-photon microscopy. We would also like to thank Dr Ichiko Kinjo and Dr Eliseo Castillo for critical reading of the manuscript.

## Author contributions

P.M. and J.L.C. and wrote the paper. P.M., J.L.C. and M.J.M. designed experiments. P.M., M.J.M., L.Y. and S.R.O. performed experiments. P.M., G.M.F., J.T., M.E.M., and J.L.C. designed approaches for data-analysis. P.M., G.M.F., J.T., J.R.B. and S.L.H. performed data analysis. All authors provided input into interpretation of data and writing of the paper.

## Additional information

**Competing interests:** The authors declare no competing financial interests.

