## [Peer Review File · Nature Communications]

Reviewers' comments:

Reviewer #1 (Remarks to the Author):

Recommendation : Major Revision. Yes, suitable if Q1-3 are adequately addressed

ROCK enables interstitial movement of T cells along the vasculature of inflamed lungs

The authors study CD8 motility in lung slices from mice recovering from LPS-induced lung-injury.

There are some nice technical points here and it's a very solid start to something potentially interesting but the analysis of the 'meat' (Figure 7/8) are too minimal and too many uncertainties remain. I'd be supportive of this for Nat. Comm. based on its potential import but as presented I don't see this as a strong addition to our understanding of migration based on the following queries:

Q1: Is this really interstitial motility being studied here or might some of it also be alveolar?

This is quite important for interpreting the inhibitor studies but the question starts to appear important earlier in the paper.

In Movie S4, the cell appears to crawl away from the marked vasculature. Similarly, the last half of the track for Movie S5 and then the un-tracked cell in Movie S6. Can the authors confirm that this is along a surface (as in 'in' the interstitium) and that the cell hasn't crossed all the way into the agarose within the filled alveolus? How is this accounted for in the analysis. The vascular anti-CD3 only shows that it is not in the vessel but don't distinguish interstitial versus intra-alveolar. Can they provide movies that reinforce that it is the former (or, more difficult, inhale anti-CD3PE and do an 'alveolar' labeling)?

To this extent, the data shown in Figure 5 is almost too good (if those movies are representative). Was there some filtering done here to only keep the 'green' segments for Figure 5D. If so, another graph should be generated that shows green versus white segments.

Overall, although there is no doubt that some portions of some tracks align with the vascular marking, it may not be possible with this approach to be sure they are not looking at hugging of the inner alveolar wall. The authors should ideally confirm or refute this with data/analysis.

Q2: How much of the turning angle analysis is a function of structure?

In Figure 3, the authors attempt to fit the movements to a variety of search patterns. They find that log-normal is the best and that neither Brownian nor Levy behavior is well supported. They also find that turning angle is well correlated but only for the first minute. However, more thought could go into the role of the convolutions of the interstitium in influencing perceived tracks. The correlation of turning angles with short time could merely be a reflection of the structure of the interstitium, no? Given the absence of flow, too, its interesting to consider why cells would ever move in a straight path in the lung unless following a chemokine gradient, perhaps laid down on the inside of the intersitium. (See possible confinement issue remaining with PTX, Q3)

Q3: Can the authors provide more robust analysis of the inhibitor results?

Beyond the import of Q1 for this, the inhibitor results are the most surprising and potentially important but there are some problems with the setups and interpretation that really lessen my enthusiasm for the paper as presented.

First, Figure 7 is baffling. "A" shows very limited change in speed (see below about whether PTX block was complete) and "C" shows very little change in turning behavior. "B" is interpreted as showing a decent change in MSD displacement but the xy intercept was allowed to float for the control and if I am looking at that data, I see a bit of a dampening of MSD (flattening toward the x axis) with time in the control, suggesting maybe that chemokine normally might confine cells. But in general, there needs more exploration of this point to make this clearer. In general, the paper would be stronger if some of the early figures were combined or supplementalized so as to spend more time analyzing this result. What are the effects of this inhibitor and the Rock inhibitor as you move toward the airway?

Additionally, the inhibitor quality should be in doubt: For Figure 7, they state they incubate "the

lungs for two hours to ensure effective PTX inhibition" but shouldn't there be a functional way to show this? It may be that all of the PTX stays trapped at the border of the slide and doesn't effectively diffuse. Given that they have very minimal effects (quite unexpected) I'd expect to see a better assay here. Two experiments come to mind. First, they can reisolate the T cells from the lung slices after PTX addition and do a transwell assay to show they are blocked. Second, they could add a gradient to the lung (CXCL12 impregnated beads) and show responsiveness. Third, they could adopt cells that were only just treated with PTX and watch them later (PTX takes about 30 minutes to deplete the Gai). Note too that there are decent chemokine blocking antibodies available and if there really is a slight slowing but a de-confinement (my interpretation of Fig 7) with chemokine loss, they should be able to add antibodies to the slices as well (anti-CXCL10/12, see above for needed controls).

Also, I would think that, unless there are other factors they are proposing here and which would then need to be demonstrated to complete this study, that the Rock inhibitor should phenocopy at least a component of the PTX/chemokine blockade; since we know/they show that this pathway is downstream of the chemokines they are proposing are important. It'd be fine if Rock inhibitor didn't have a phenotype but chemokine did but the reverse in this situation seems odd. I suggest again that they check their PTX efficacy and update that result or provide an alternative explanation for the Rock effect. At present it is otherwise a rather isolated observation that should be advanced (what molecule drives Rock-dependent effects if not chemokine?).

Other queries

Figure 6 would appear to have limited novelty and could at very least be put in supplemental. The relevance of CD8+ T cells is not well established in lung injury when compared to CD4 T cells. As the authors show CD8s are clearly recruited to the lung in lung injury, but to my knowledge no one has demonstrated that they play an important role in the pathology of ARDS compared to CD4 subsets such as Tregs and Th17s. This may significantly lowers the impact of this study when compared to if it was done on CD4 subsets relevant to ARDS, or if it had first demonstrated a key function for CD8s in ARDS. Can the authors at least provide justification for why this particular biology matters or if CD4s are the same?

Other findings (no queries here, just for reflection)

In Figure 1, the authors show that activated CD8 T cells extravasate into the injured lung, out of the vasculature. This is quite good data and the only caveat regarding interpretation is in Q1.

In Figure 2, the authors demonstrate variation in speeds, highlighted by differing the integration time. They also show data suggestive of a sinusoidal pattern of migration speeds, suggesting periodic oscillations of fast/slow migration.

In Figure 3, the authors attempt to fit the movements to a variety of search patterns. They find that log-normal is the best and that neither Brownian nor Levy behavior is well supported. They also find that turning angle is well correlated but only for the first minute. However, more thought could go into the role of the convolutions of the interstitium in influencing perceived tracks. The correlation of turning angles with short time could merely be a reflection of the structure of the interstitium, no? Given the absence of flow, too, its interesting to consider why cells would ever move in a straight path in the lung unless following a chemokine gradient, perhaps laid down on the inside of the intersitium.

In Figure 4, the authors do a highly synthetic exercise to grab track segments to create 'synthetic tracks' and then compare these to the actual. Over a number of observations and modeling exercises, they provide convincing data that cells do indeed oscillate in behavior over time.

Reviewer #2 (Remarks to the Author):

In this study, Mrass et al. examined CD8⁺ effector T cells motility using a LPS-induced acute lung injury model. Following adoptive transfer of polyclonal CD8⁺T cells activated in vitro into mice treated LPS intranasally. They found that transferred T cells efficiently migrated into lung tissues. Two-photon imaging of explanted lungs up to two hours showed characteristic behaviors of transferred T cells. Computational analysis of manually tracked T cells in lung tissues revealed that lung-infiltrating T cells had a tendency of intermittent repeats of fast/directional and slow/confined migration. The fast/directional migration appeared to associated with the vascular structure. The authors tried to dissect the molecular background to explain lung-infiltrating T cells migration behaviors. Pertussis toxin modestly decreased velocities of lung-infiltrating T cells while the directionality was maintained. In contrast, ROCK inhibitor greatly diminished both velocities and directionality. Based on these findings, the authors speculate that T cells squeezed through preformed tissue channels presumably through a Rho-ROCK-F-actin dependent amoeboid migration.

The excellent part of this study is statistical analyses of motility of lung-infiltrating T cells. By comparing appropriately randomized processes, they revealed a tendency of intermittent repeats of fast/directional and slow/confined migration. The experiments using PTX or ROCK inhibitors revealed ROCK-dependent directional migration was dominant whereas inhibition of Gi signaling modestly reduced velocities. Unfortunately, biological processes behind infiltrating T cell behaviors were poorly examined. The authors suggest directional T cell movement along preformed channels associated with the vasculature, which was not characterized in details. It remains unknown that what triggers Rho-ROCK signaling in inflamed lung tissues. The involvement of integrins was not addressed. What is the biological significance of the intermittent T cell migration and what roles CD8⁺ effector T cells play by doing so in a LPS-induced lung injury model? Although the statistical characterization of T cell migration patterns in inflamed lungs using two-photon microscopy is interesting, data obtained using explanted lungs might include artifacts (altered lung tissue architecture due to loss of air and blood flow, for example), and should be validated by comparing using in vivo imaging methods. Overall, I think that the findings described in this study are potentially interesting, but not enough to advance our conceptual understanding of interstitial migration that was reported before.

Manuscript NCOMMS-16-19169-T

Former title: ROCK enables interstitial movement of T cells along the vasculature of inflamed lungs

New title: Rho-associated protein kinase (ROCK) regulates the intermittent mode of interstitial T cell migration in inflamed lung

We appreciate the reviewers' insightful critiques. Based on the reviewer's comments and revisions, we have adjusted the title. We have addressed all the concerns in our point-by-point response below and hope that the editor and reviewers will find the work appropriate for publication in *Nature Communications*. Please note that new text in the revision is indicated by yellow highlights.

Reviewer #1:

Q1: Is this really interstitial motility being studied here or might some of it also be alveolar?

This is quite important for interpreting the inhibitor studies but the question starts to appear important earlier in the paper.

In Movie S4, the cell appears to crawl away from the marked vasculature. Similarly, the last half of the track for Movie S5 and then the un-tracked cell in Movie S6. Can the authors confirm that this is along a surface (as in 'in' the interstitium) and that the cell hasn't crossed all the way into the agarose within the filled alveolus? How is this accounted for in the analysis. The vascular anti-CD3 only shows that it is not in the vessel but don't distinguish interstitial versus intra-alveolar. Can they provide movies that reinforce that it is the former (or, more difficult, inhale anti-CD3PE and do an 'alveolar' labeling)?

To this extent, the data shown in Figure 5 is almost too good (if those movies are representative). Was there some filtering done here to only keep the 'green' segments for Figure 5D. If so, another graph should be generated that shows green versus white segments.

Overall, although there is no doubt that some portions of some tracks align with the vascular marking, it may not be possible with this approach to be sure they are not looking at hugging of the inner alveolar wall. The authors should ideally confirm or refute this with data/analysis.

We previously provided data (Fig 1) that demonstrated that the effector T cells we image are not intravascular. Based on the reviewer's suggestion, we have now performed intranasal anti-CD3 (T cell) staining in combination with intravascular staining (now Fig 1C). This data shows that approx. 80% of T cells are negative for intravascular and intranasal anti-CD3 stain, demonstrating that T cells are extravascular and not in the airway space exposed to intranasal staining. In several experiments, we consistently demonstrate that 70-90% of T cells are extravascular and extra-alveolar. This is also stated in new text on page 5 (new text indicated with yellow highlights)

With regards to the question whether filtering was done in previous Fig 5D, no filtering was done.

Q2: How much of the turning angle analysis is a function of structure?

In Figure 3, the authors attempt to fit the movements to a variety of search patterns. They find that log-normal is the best and that neither Brownian nor Levy behavior is well supported. They also find that turning angle is well correlated but only for the first minute. However, more thought could go into the role of the convolutions of the interstitium in influencing perceived tracks. The correlation of turning angles with short time could merely be a reflection of the structure of the interstitium, no? Given the absence of flow, too, its interesting to consider why cells would ever move in a straight path in the lung unless following a chemokine gradient, perhaps laid down on the inside of the intersitium. (See possible confinement issue remaining with PTX, Q3)

The reviewer suggests that the turning angles taken by the T cells may reflect structural determinants in the interstitium. We agree and our analysis of vascular guidance of T cell motion suggests exactly this. In addition, we have now performed additional analysis that agrees with the reviewer's hypothesis. In the new Fig 5E, we now look at T cells that move in similar areas within the same field. We find that T cells in similar areas actually move in similar trajectories, suggesting that the structural components of the lung can help T cells move in a specific trajectory. This is now added on page 10.

Q3: Can the authors provide more robust analysis of the inhibitor results?

Beyond the import of Q1 for this, the inhibitor results are the most surprising and potentially important but there are some problems with the setups and interpretation that really lessen my enthusiasm for the paper as presented.

First, Figure 7 is baffling. "A" shows very limited change in speed (see below about whether PTX block was complete) and "C" shows very little change in turning behavior. "B" is interpreted as showing a decent change in MSD displacement but the xy intercept was allowed to float for the control and if I am looking at that data, I see a bit of a dampening of MSD (flattening toward the x axis) with time in the control, suggesting maybe that chemokine normally might confine cells. But in general, there needs more exploration of this point to make this clearer. In general, the paper would be stronger if some of the early figures were combined or supplementalized so as to spend more time analyzing this result. What are the effects of this inhibitor and the Rock inhibitor as you move toward the airway?

Additionally, the inhibitor quality should be in doubt: For Figure 7, they state they incubate "the lungs for two hours to ensure effective PTX inhibition" but shouldn't there be a functional way to show this? It may be that all of the PTX stays trapped at the border of the slide and doesn't effectively diffuse. Given that they have very minimal effects (quite unexpected) I'd expect to see a better assay here. Two experiments come to mind. First, they can reisolate the T cells from the lung slices after PTX addition and do a transwell assay to show they are blocked. Second, they could add a gradient to the lung (CXCL12 impregnated beads) and show responsiveness. Third, they could adopt cells that were only just treated with PTX and watch them later (PTX takes about 30 minutes to deplete the Gai). Note too that there are decent chemokine blocking antibodies available and if there really is a slight slowing but a de-confinement (my interpretation of Fig 7) with chemokine loss, they should be able to add antibodies to the slices as well (anti-CXCL10/12, see above for needed controls).

Also, I would think that, unless there are other factors they are proposing here and which would then need to be demonstrated to complete this study, that the Rock inhibitor should phenocopy at least a component of the PTX/chemokine blockade; since we know/they show that this pathway is downstream of the chemokines they are proposing are important. It'd be fine if Rock inhibitor didn't have a phenotype but chemokine did but the reverse in this situation seems odd. I suggest again that they check their PTX efficacy and update that result or provide an alternative explanation for the Rock effect. At present it is otherwise a rather isolated observation that should be advanced (what molecule drives Rock-dependent effects if not chemokine?).

We apologize if our previous description of previous Fig 7 (now Fig 6) was misleading. It is very clear from the speed and MSD data that PTX inhibited T cell speed and displacement at similar levels as previously published in other organs (Harris et al. 2012, see additional citations in text). In former Fig 7A (now 6A), speed is decreased by from 2.4 $\mu\text{m}/\text{min}$ in untreated cells to 2.0 $\mu\text{m}/\text{min}$ in PTX inhibited cells. More importantly, the motility coefficient of T cells decreased by 50% from untreated to PTX treated T cells (from 6.8 $\mu\text{m}^2/\text{min}$ to 3.6 $\mu\text{m}^2/\text{min}$). In former 7B, displacement shows the slope of the MSD is significantly lower, showing less displacement in the presence of PTX. We have now adjusted the text to reflect this (see page 11). The reviewer is correct that the MSD may be slightly flatter for the control, but we feel that the data is not sufficient to support the conclusion that chemokine confines cells based solely on this analysis.

The reviewer suggests that *“some of the early figures were combined or supplementalized so as to spend more time analyzing this result.”*

We have now done so (see new Supplemental Fig S4).

The reviewer also asks: *“What are the effects of this inhibitor and the Rock inhibitor as you move toward the airway?”*

Unfortunately, because the ROCK inhibitor leads to a dramatic decrease in motility, we cannot easily analyze whether T cells in the presence of ROCK inhibitor change close to airways.

Additionally, the inhibitor quality should be in doubt: For Figure 7, they state they incubate “the lungs for two hours to ensure effective PTX inhibition” but shouldn't there be a functional way to show this?

The reviewer asks that we demonstrate that PTX is functionally blocking chemokine signaling. We believe that the decrease in speed and displacement clearly shows that PTX blocks T cell motion. The only aspect that PTX does not inhibit is the intermittent mode of motion. This is consistent with previous findings in Harris et al. 2012 in which T cell speed but not motility pattern was not changed in infected brain. In addition, we have now added an in vitro confirmation of PTX block (shown in Supplemental Fig S4). T cells incubated with PTX for 2 hours under the same conditions as the lung explants show a decrease in transwell migration to a similar degree as seen in explanted tissue.

The reviewer also suggests that PTX may be trapped outside the tissue. We have done control experiments that demonstrate that anti-CD3 antibodies can diffuse and stop T cell motion into intact lung using the same perfusion setup that we use for PTX (see Fig 1 within response).

Anti-CD3 antibody treatment led to a complete inhibition of T cell motion in explanted lung tissue. As antibodies are large molecules, we expect that PTX should diffuse into lung tissue similar to anti-CD3 antibodies (antibodies: ~150kDa; PTX: 117kDa).

The reviewer asks whether additional reisolation experiments can be done. These are not technically feasible as the extraction of T cells from tissues take multiple hours so extraction would prevent analysis of PTX inhibition within the native tissue.

We feel that using antibodies to individual chemokines may not fully block all the potential chemokines that may act and that PTX is a more complete inhibition of chemokine receptor signaling.

Our data agrees with the reviewer that the ROCK inhibitor does phenocopy PTX treatment at least in the decrease in both speed and displacement (MSD). The ROCK inhibitor and PTX are not actually reversed from PTX, but instead, ROCK inhibition blocks to a greater degree than PTX (compare Fig 6 and 7). This also agrees with our newly added in vitro inhibition of transwell migration (supplemental fig S4). We now show that both PTX and ROCK inhibitor block transwell migration, but ROCK inhibits to a greater degree than PTX, even in baseline migration in the absence of specific chemokine receptor signaling. These data show that while ROCK likely acts downstream of chemokine receptor G α i signaling, ROCK has a role in addition to G α i and thus, inhibition of ROCK shows a more dramatic decrease in T cell motion.

Responses to minor queries of Reviewer #1

Figure 6 would appear to have limited novelty and could at very least be put in supplemental.

We have now changed previous Fig 6 into new Supplemental Fig S4.

The relevance of CD8+ T cells is not well established in lung injury.

Multiple papers have clearly shown a pathogenic role for CD8 T cells in lung injury models resulting from infections (Connors TJ et al., 2016; Xu et al., 2004; Van den Steen et al., 2010). In addition, several papers directly demonstrate the role of CD8 in lung ARDs (Li GG et al., 2016; Li N et al., 2016). We have now added some of these to the text as citations for the potential role for CD8 T cells in lung injury (page 4).

Response to Reviewer #2 queries:

We appreciate that the reviewer felt that *“The excellent part of this study is statistical analyses of motility of lung-infiltrating T cells.” Unfortunately, biological processes behind infiltrating T cell behaviors were poorly examined.*

The most significant concern of the reviewer was the biological process of intermittent migratory behavior. We have now added a model that suggests a potential biological role for intermittent behavior in new Fig 8. The model strongly suggests that the straight portion of intermittent migration allows T cells to cover a large area but the confinement gives T cells the opportunity to make longer contacts with potential targets.

The authors suggest directional T cell movement along preformed channels associated with the vasculature, which was not characterized in details.

We have now added Fig 5E which shows that distinct T cells that move in the same region move with similar trajectories that align with vasculature (see response to reviewer #1 Q2). This further supports our conclusion that vasculature may provide structural guidance for T cell motion.

It remains unknown that what triggers Rho-ROCK signaling in inflamed lung tissues.

We show that chemokine receptor can induce Rho activation. However, our results also demonstrate that ROCK activity can be active at baseline, simply in serum, as demonstrated by Supplemental Fig S4 where ROCK inhibitor can block baseline transwell migration, even in the absence of exogenously added chemokine. Thus, it is likely that multiple signals in vivo including but not limited to chemokine signaling can lead to ROCK activation.

The involvement of integrins was not addressed.

While the role of integrins is of interest, we feel that the role of integrins is outside the scope of this particular study, which focuses on the role of chemokines and intrinsic signaling molecule ROCK in driving T cell motion in lung.

What is the biological significance of the intermittent T cell migration and what roles CD8+ effector T cells play by doing so in a LPS-induced lung injury model?

Please see response above regarding biological significance as well as to Reviewer #1 minor query above.

T cell migration patterns in inflamed lungs using two-photon microscopy is interesting, data obtained using explanted lungs might include artifacts (altered lung tissue architecture due to loss of air and blood flow, for example), and should be validated by comparing using in vivo imaging methods.

While we agree with reviewer #2 that some aspects of lung physiology depend on blood and air flow that are maintained during intravital imaging, T cell motion does not appear to be dependent on blood and air flow in many tissues, including lung, lymph nodes and brain. Multiple publications study interstitial motility in lung, lymph nodes, and brain. In lung, the following papers all use explanted tissue to track T cell motion in lung: Thornton et al. JEM 2012; Matheu et al. PONE 2013; Torabi-Parisi JI Germain 2014; Halle et al. JEM 2009. T cell motion using explanted tissue has been done in brain (Harris et al. Nature 2012), lymph node (Miller et al. PNAS 2003) and tumor (Mrass et al. JEM 2006). Even in side by side studies of T cell motion in intravital and explanted organs, no differences have been clearly documented.

A few papers have used intravital imaging to image immune responses in lung, and most of these studies use both explant and intravital side by side. Fiole et al. (Infection and Immunity 2014) visualizing anthrax in lung uses slices and intravital both interchangeably. In fact, from direct side by side comparisons of T cell motion in lung in Thornton 2012, the authors state that there are no observed differences between explanted lung slices and intravital imaging in either dendritic cell or T cell behavior (Thornton JEM 2012). In addition, because of the need for precise T cell tracking in our detailed quantitative analyses, we believe that intravital imaging would introduce additional motion artifacts that would inhibit our ability to track individual T cells. Thus, we believe that our use of explanted lung in tracking T cells is well justified.

References cited in response:

Van den Steen PE, Geurts N, Deroost K, Van Aelst I, Verhenne S, Heremans H, Van Damme J, Opendakker G. 2016 Immunopathology and dexamethasone therapy in a new model for malaria-associated acute respiratory distress syndrome. Am J Respir Crit Care Med. 2010 May 1;181(9):957-68.

Connors TJ, Ravindranath TM, Bickham KL, Gordon CL, Zhang F, Levin B, Baird JS, Farber DL. 2016. Airway CD8(+) T Cells Are Associated with Lung Injury during Infant Viral Respiratory Tract Infection. Am J Respir Cell Mol Biol. 2016 Jun;54(6):822-30. doi: 10.1165/rcmb.2015-0297OC.

Xu L, Yoon H, Zhao MQ, Liu J, Ramana CV, Enelow RI. 2004. Cutting edge: pulmonary immunopathology mediated by antigen-specific expression of TNF-alpha by antiviral CD8+ T cells. J Immunol. 2004 Jul 15;173(2):721-5.

Li GG, Cao YH, Run Y, Xu RX, Zheng ZD. 2016. Inhibition of CD8+ T cells and elimination of myeloid cells by CD4+ Foxp3- T regulatory type 1 cells in acute respiratory distress syndrome. *Clin Exp Pharmacol Physiol*. 2016 Dec;43(12):1191-1198. doi: 10.1111/1440-1681.12656.

Li Nie, Wei Wu, Zhibing Lu, Gangyan Zhu, and Juan Liu 2016. CXCR3 May Help Regulate the Inflammatory Response in Acute Lung Injury via a Pathway Modulated by IL-10 Secreted by CD8 + CD122+ Regulatory T Cells. *Inflammation*. 2016; 39: 526–533.

Miller, M. J., S. H. Wei, M. D. Cahalan, and I. Parker. 2003. Autonomous T cell trafficking examined in vivo with intravital two-photon microscopy. *Proc Natl Acad Sci U S A* 100: 2604-2609.

Mrass, P., H. Takano, L. G. Ng, S. Daxini, M. O. Lasaro, A. Iparraguirre, L. L. Cavanagh, U. H. von Andrian, H. C. Ertl, P. G. Haydon, and W. Weninger. 2006. Random migration precedes stable target cell interactions of tumor-infiltrating T cells. *J Exp Med* 203: 2749-2761.

Looney, M. R., E. E. Thornton, D. Sen, W. J. Lamm, R. W. Glenny, and M. F. Krummel. 2011. Stabilized imaging of immune surveillance in the mouse lung. *Nat Methods* 8: 91-96.

Thornton, E. E., M. R. Looney, O. Bose, D. Sen, D. Sheppard, R. Locksley, X. Huang, and M. F. Krummel. 2012. Spatiotemporally separated antigen uptake by alveolar dendritic cells and airway presentation to T cells in the lung. *J Exp Med* 209: 1183-1199.

Halle, S., H. C. Dujardin, N. Bakocevic, H. Fleige, H. Danzer, S. Willenzon, Y. Suezter, G. Hammerling, N. Garbi, G. Sutter, T. Worbs, and R. Forster. 2009. Induced bronchus-associated lymphoid tissue serves as a general priming site for T cells and is maintained by dendritic cells. *J Exp Med* 206: 2593-2601.

Matheu, M. P., J. R. Teijaro, K. B. Walsh, M. L. Greenberg, D. Marsolais, I. Parker, H. Rosen, M. B. Oldstone, and M. D. Cahalan. 2013. Three phases of CD8 T cell response in the lung following H1N1 influenza infection and sphingosine 1 phosphate agonist therapy. *PLoS One* 8: e58033.

Torabi-Parizi, P., N. Vrisekoop, W. Kastenmuller, M. Y. Gerner, J. G. Egen, and R. N. Germain. 2014. Pathogen-related differences in the abundance of presented antigen are reflected in CD4+ T cell dynamic behavior and effector function in the lung. *J Immunol* 192: 1651-1660.

Daniel Fiole, Pierre Deman, Yannick Trescos, Jean-François Mayol, Jacques Mathieu, Jean-Claude Vial, Julien Douady, and Jean-Nicolas Tournier. 2014. Two-Photon Intravital Imaging of Lungs during Anthrax Infection Reveals Long-Lasting Macrophage-Dendritic Cell Contacts. *Infect Immun*. 2014 Feb; 82(2): 864–872.

Reviewers' comments:

Reviewer #2 (Remarks to the Author):

In the revised manuscript, the authors did additional experiments and also provided new evidence from simulation study to support the intermittent migration controlled by Rho-ROCK had an advantage for target contact duration. Although the target for CD8 T cells in the LPS model is unclear, this finding would be meaningful for CD8 T cell functions.

The authors showed that spontaneous T-cell migration through 3 μm pores was severely reduced by the ROCK inhibitor. This result suggests that Rho-ROCK rather than Gi signaling stimulate confined motility. The authors also showed ERK activation in the same figure (Fig S4C). This is rather redundant data since it is already known that ERK activity is increased by chemokines. Or is there any evidence that ERK trigger ROCK activation, or ERK inhibitors block T cells motility in the LPS-injured lung?

Regarding the explant vs in vivo imaging, the studies of lymph nodes showed these were comparable in interstitial migration. The cited study (Thornton E.E. et al) did not actually compare T cell interstitial migration in sliced lung tissues and in vivo live lung imaging. The results that the straight, fast movement often aligned the vascular structures, as shown in this study, suggest the increased blood flow could affect vascular structures and hence T cell migration behaviors in associated confined space. I feel that this issue might deserve careful side-by-side observation and analysis in LPS-injured models, and the insight of a possible contribution of blood flow to confined T cell motility and intermittent migration as well as clarification of target finding functions would really advance our understanding in inflammatory lung pathology.

Overall, as the first round review, I think that the authors' analytical methods are excellent, as also indicated in new simulation study, but biological ones are still limited.

Response to reviewer

NCOMMS-16-19169A

Rho-associated protein kinase (ROCK) regulates the intermittent mode of interstitial T cell migration in inflamed lung

Mrass et al.

We appreciate the opportunity to resubmit our manuscript. In the previous round of review, the reviewer asked that we determine whether the type of intermittent motility is physiologically relevant by analyzing T cell motility by intravital two photon microscopy, citing the uniqueness of the lung with regards to the air exchange and blood flow.

My point is whether the T cell behaviors observed in inflamed lung explants are physiologically relevant or not. This issue could be approached in two ways; by intravital lung imaging in order to confirm that this finding holds true in vivo, and/or by clarification of its biological effect on inflamed lungs. I feel that either approach with reasonable findings would be acceptable.

To address this remaining concern, we have now performed intravital imaging of inflamed lung in the acute injury model. As suggested by the reviewer, we established a collaboration with a leader in the field of intravital imaging, Dr. Mark Miller of Washington University in St. Louis, who is an author on one of the papers cited by the reviewer (In vivo two-photon imaging reveals monocyte-dependent neutrophil extravasation during pulmonary inflammation. 2010. *PNAS*). Dr. Miller is one of only a few labs in the world who are performing intravital two photon microscopy in inflamed lung.

In collaboration with the Miller group, we have now imaged T cell motility patterns in inflamed lungs of living mice with acute lung injury with intact air exchange and blood flow. We find that intravital imaging of T cell movement in inflamed lung show a similar pattern of intermittent motion as explanted lung in individual T cells (new **Figure 4G**) as well as in the full population of T cells that we analyzed (new **Figure 4I**). We also find that the difference in speed over longer periods of imaging are maintained (new **Figure 4H**). In this revision, we have added new text to the results as well as discussion to reflect our new results (text highlighted in yellow), confirming that intermittent T cell motion occurs in the lung in a physiologically intact model of ALI. Dr. Miller and his group are now co-authors on this revised manuscript to reflect their contribution to intravital imaging. This new intravital imaging results confirm our novel finding that intermittent migration of T cells is physiologically relevant.

We have also included a supplemental movie from intravital imaging showing intermittent T cell motion (new **Supplemental Movie S4**). We have not included control movies demonstrating blood flow from intravital imaging but we are happy to provide additional control movies to the reviewer if requested.

We hope that with these additional controls, the editor and reviewer will find our manuscript suitable for publication in *Nature Communications*. Below is the full reviewer's response to our previous rebuttal for your reference. Thank you, Judy Cannon

REVIEWER RESPONSE TO PREDICTIVE REBUTTAL

I have carefully read the rebuttal letter.

My point is whether the T cell behaviors observed in inflamed lung explants are physiologically relevant or not. This issue could be approached in two ways; by intravital lung imaging in order to confirm that this finding holds true in vivo, and/or by clarification of its biological effect on inflamed lungs. I feel that either approach with reasonable findings would be acceptable. Disappointingly, however, the authors declined none of experiments to address this issue. The assumption that T cell motilities in vivo and ex vivo are comparable based on other organs would be inappropriate, as the lung is a unique organ; the parenchyma is composed of thin alveolar walls with rich air and blood flows. It would be rather counterintuitive that confined T cell motilities are not affected by their pressure. Although dendritic cells in the lung appeared similar ex vivo and in vivo, T cells are much faster (as indicated in this study) and should be studied separately. The experimental method of intravital imaging of the inflamed lungs was first published in 2010. Since then, unique leukocytes and cancer cells motilities were reported, which are often associated with the vasculature (some references are listed below). To my knowledge, the apparatus and methods for intravital lung imaging has been widely known. When it is difficult to get ones, the authors should seek for collaborations. As I mentioned above, the authors should clarify the biological effect on inflamed lungs, alternatively. I think that at least either approach should be taken.

Overall, I think that this study is rather analytical as it is (which is excellent), but insufficient for a conceptual advance, since the most findings of this study were already reported in other organs.

REVIEWERS' COMMENTS:

Reviewer #2 (Remarks to the Author):

The revised paper now includes live imaging of T cell migration using 2-P microscopy, which also showed intermittent migration patterns. This information greatly strengthens physiological relevance of the data obtained *ex vivo*. I recommend the revised paper for publication in Nature Communications.

Manuscript NCOMMS-16-19169B

Point by point response

We are pleased that reviewer #2 found our response satisfactory and recommends our paper for publication. In response to reviewer #2's comment on our response to reviewer #1 regarding chemokine signaling to ROCK, we have now included a sentence in the discussion stating clearly that we do not have a clear mechanism of ROCK activation by chemokine in vivo (indicated by tracked changes). Below is the original comments by reviewer #2. We hope that this satisfies all reviewer requirements for publication in *Nature Communications*.

Reviewer #2 (Remarks to the Author):

The revised paper now includes live imaging of T cell migration using 2-P microscopy, which also showed intermittent migration patterns. This information greatly strengthens physiological relevance of the data obtained ex vivo. I recommend the revised paper for publication in Nature Communications.

Please also note that we asked reviewer 2 to comment on your response to reviewer 1 from last round, as reviewer 1 report was not particularly informative. Reviewer 2 tells us that your responses to reviewer 1 are satisfactory in most cases, but thinks it would have been better to have provided some Gq inhibitor studies or similar to gain a greater understanding of what regulates ROCK signalling in this context. As such we also ask that you add a clear caveat in discussion outlining that you do not have a clear mechanism of this induction.